

**The hydrodynamic and environmental characteristics of tributary bay**
**influenced by backwater jacking and intrusion of main reservoir**
Xintong Li[1], Bing Liu[2], Yuanming Wang[1], Yongan Yang[3], Ruifeng Liang[1]*, Fangjun
Peng[1], Shudan Xue[1], Zaixiang Zhu[1], Kefeng Li[1]
*[1] State Key Laboratory of Hydraulics and Mountain River Engineering, Sichuan University, Chengdu 610065, China*
*[2] Emergency Response Centre, Ecology and Environment Bureau of Suining, Suining 629000, China*
*[3] Environmental Monitoring Centre, Ecology and Environment Bureau of Suining, Suining 629000, China*
**Abstract.** The construction of large reservoirs results in the formation of tributary
bays, and tributary bays are inevitably influenced by the backwater jacking and
intrusion of the main reservoir. The hydrodynamic conditions and the environmental
factors of tributary bays exhibit complex distribution characteristics and
eutrophication occur frequently. Thus, exploring the distribution and evolution of the
hydrodynamic and water environment characteristics of tributary bays in response to
backwater jacking and intrusion is the key to solving eutrophication and other
problems relevant to water environment. In this paper, a typical tributary bay (Tangxi
River) of the Three Gorges Reservoir (TGR) was selected to study the hydrodynamic
and environmental characteristics of the tributary bay influenced by the jacking and
intrusion of the main reservoir. The flow field, water temperature and water quality of
the Tangxi River were simulated using the hydrodynamic and quality model
CE-QUAL-W2, and the eutrophication status of the tributary bay was also evaluated.
The results showed that the main reservoir had different effects on its tributary bay in





each month. The tributary bay was mainly affected by backwater jacking of the main
reservoir when the water level dropped and by intrusion of the main reservoir when
the water level rose. An obvious quality concentration boundary existed in the
tributary bay, which was basically consistent with the regional boundary in the flow
field. The flow field and water quality on both sides of the boundary were quite
different. The results of this study can help us figure out how the backwater jacking
and intrusion of the main reservoir influence the hydrodynamic and water
environment characteristics of the tributary bay and provide guidance for water
environment protection in the tributary bays.
**Keywords:** tributary bay, main reservoir, backwater jacking, intrusion, hydrodynamic
conditions, environmental factors
**1 Introduction**
The functions of water conservancy and hydropower projects include power
generation, flood control, irrigation and shipping, which play an important role in
human social life (Deng and Bai, 2016; Zhang, 2014; Peng, 2014). In recent years,
with the construction of the Yangtze River Economic Belt and urban agglomeration of
China, a large number of high dams, with heights of over 200 m or even 300 m, have
been planned or completed in the middle and upper reaches of the Yangtze River to
meet the increasing energy demand (Zhou et al., 2013). However, these dams block
the fish migration routes between upstream and downstream regions (Oldani and
Claudio, 2002; Ziv et al., 2012) and change the fish communities (Gao et al., 2010).



In the flood season, flood discharge produces water that is supersaturated in dissolved
gas in the downstream river channel (Feng et al., 2014; Lu et al., 2011; Wang et al.,
2011; McGrath, 2006). In the reservoir area, the elevated water level produces a much
slower water velocity, which results in sediment deposition, eutrophication, and
stratification in terms of water temperature and water quality (Zhu, 2017; Wu, 2013;
Zhang et al., 2011).

Backwater extends to some tributaries after the construction of dammed-river

reservoirs, which causes the water depth to increase and the water velocity to slow in
these tributaries, thus formed the water areas similar to lakes, and were known as
tributary bay (Yu et al., 2013). Backwater areas represent the connection between
different habitats in the main stream and the tributary and are also an important
location for physical, chemical and biological exchanges between adjacent habitats
(Zhang et al., 2010). After the impoundment of a reservoir, the hydrodynamic
conditions and the environmental factors (water temperature, water quality, etc.) of
the tributaries in the reservoir area are affected by the main stream and exhibit
complex distribution characteristics (Xiong et al., 2013). A tributary bay is always
influenced by backwater jacking and intrusion with the rise of the water level of the
main reservoir because such changes induce changes in the hydrodynamic conditions
in the tributary bay. The velocity of water in the horizontal direction becomes uneven,
and the velocity on the side near the confluence is obviously higher than that on the
other side. The flow field distribution tends to gradually change with increasing



distance from the confluence (Yin et al., 2013). The water level of a reservoir changes
constantly to meet multiple requirements, which results in changes in water
temperature and water environment in tributary bays. Existing studies have shown
that water level fluctuation has become a major cause of recent eutrophication and
pollution problems in the TGR, particularly within its tributary backwaters (Holbach
et al., 2015). After the impoundment of reservoirs, eutrophication and
eutrophication-related problems often occur in tributary bays due to changes in
nutrient patterns (Yang et al., 2010; Liu et al., 2012; Ran et al., 2019). Therefore,
exploring the distribution and evolution of the hydrodynamic and water environment
characteristics of tributary bays in response to backwater jacking and intrusion of the
main reservoir is the key to solving eutrophication problems.

Many recent studies have paid attention to the deterioration of the water

environment in tributary bays. In response to the operation of cascade reservoirs, a
series of profound geological, morphological, ecological, and biogeochemical
responses will appear in the estuary, delta, and coastal sea of Yangtze River
subaqueous delta (Hu et al., 2009). Some scholars have found that the water quality of
the TGR was relatively stable before and after impoundment but that the water quality
of tributary bays deteriorated, resulting in frequent algal blooms (Liu et al., 2016; Zou
and Zhai, 2016; Cai and Hu, 2006). Changes in the vertical mixing of layers driven by
stratified density currents were the key factor in the formation of algal blooms (Tang
et al., 2016; Zhang et al., 2015). Through isotopic measurements in the Xiangxi River



or other tributaries of the TGR, it has been found that the nutrients in tributary bays
did not originate solely in the tributary basins but instead were mainly from the main
stream of the Yangtze River and that the nutrient levels were affected by constantly
changing hydrodynamic conditions across seasons (Holbach et al., 2014; Yang et al,
2018; Zheng et al., 2016). Some scholars found that a rise in the water level may lead
either to a rise in the chlorophyll content or to a decline in the chlorophyll content,
depending on the water cycle mode in the tributary (Ji et al.,2017). The present
studies have paid considerable attention to changes in hydrodynamic characteristics
and the deterioration of the water environment in the tributaries but have not
considered the influence of the main reservoir. There are few systematic studies on the
variation in the hydrodynamic and water environment characteristics of tributary bays
influenced by the backwater jacking and intrusion of the main reservoir. How the
operation of the main reservoir affects the tributary bays, how the hydrodynamic
forces and water environment of the tributary bays respond to the backwater jacking
and intrusion of the main reservoir, what controls the water environment of the
tributary bay? These questions are not yet clear.

The Tangxi River, a tributary in the upper reaches of the Yangtze River, was

selected as the focus of this study. The hydrodynamic and water environmental
characteristics of the Tangxi River have inevitably been affected by the backwater
jacking and intrusion of the TGR in recent years. Based on the collection and analysis
of basic data, we simulated the flow field, water temperature, and water quality of the



Tangxi River using the hydrodynamic and quality model CE-QUAL-W2. Then, we
evaluated the eutrophication status of the tributary bay and systematically identified
the influence of the backwater jacking and intrusion of the main reservoir on the
tributary bay. The results of this study can help us to figure out how the backwater
jacking and intrusion of the main reservoir influenced the hydrodynamic and water
environment characteristics of the tributary bay and provide guidance for water
environment protection in the tributary bays.
**2 Materials and methods**
**2.1 Research area**
The main stream of the Yangtze River has a total length of approximately 6300 km
and a drainage area of approximately 1.8 million km$^2$. The reach between Yichang
City and Hubei Yibin City in Sichuan is considered the upper reaches of the Yangtze
River, which has a length of 1045 km and a natural drop of 220 m. The drainage area
of the upper Yangtze River is 527000 km$^2$, and its average annual flow is 14300 m$^3$/s
(Fan, 2007).

The Tangxi River is a first-order tributary of the upper Yangtze River and has a

total length of 104 km, a drainage area of 1707 km$^2$ and an average annual flow of
57.2 m$^3$/s. After the completion of the TGR, the Tangxi River became a tributary bay
of the TGR. In this paper, the 42.6 km long reach of the Tangxi River affected by the
backwater jacking and intrusion of the TGR was selected as the study area (Fig. 1).

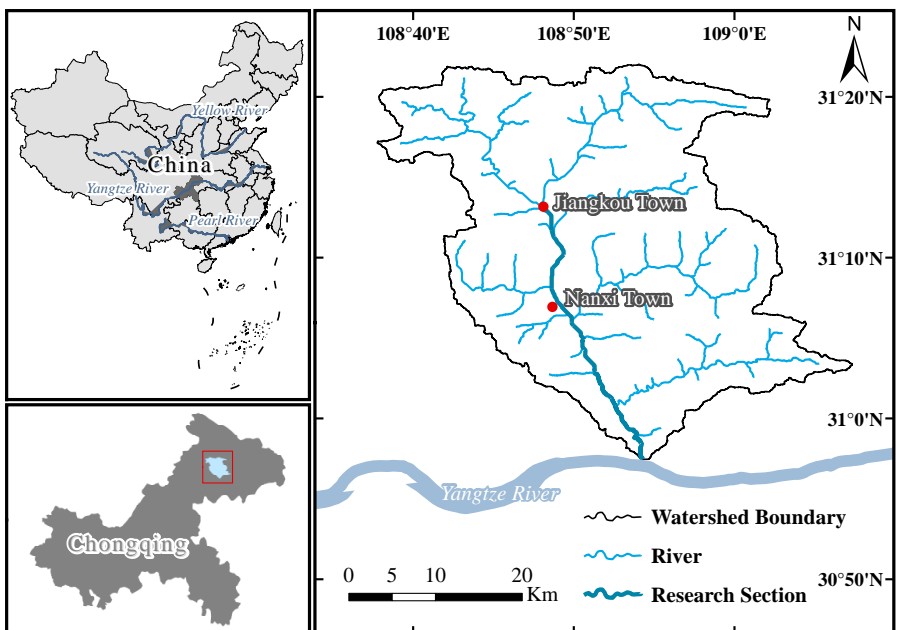

**Fig. 1.** Research area and hydrologic system of the Tangxi River Basin.

**2.2 Numerical simulation of hydrodynamic and environmental factors in the tributary bay**

**2.2.1 Mathematical model**

The vertical two-dimensional model CE-QUAL-W2 with average width was adopted for the calculation of the hydrodynamic conditions, water temperature and water quality in the tributary bay (Thomas and Scott, 2008). This model performs well in computing the velocity, the intrusion layer at the plunge point, and the travel distance of the density current (Long et al., 2019), and many scholars have obtained good results by using this model to simulate the hydrodynamics, water temperature and water quality of reservoirs and lakes (Debele et al., 2008; Noori, 2015; Long et al.,



2018). The model is solved by coupling governing equations, a transport equation and
a surface heat exchange equation.
The governing equations of the model are listed as follows.
The continuity equation:
$\frac{\partial UB}{\partial x} + \frac{\partial WB}{\partial z} = qB$ (1)
The x-momentum equation:
$\frac{\partial UB}{\partial t} + \frac{\partial UUB}{\partial x} + \frac{\partial WUB}{\partial z} = gB \sin\alpha - \frac{B}{\rho}\frac{\partial P}{\partial x} + \frac{1}{\rho}\frac{\partial B\tau_{xx}}{\partial x} + \frac{1}{\rho}\frac{\partial B\tau_{xz}}{\partial z}$ (2)
The z-momentum equation:
$\frac{1}{\rho}\frac{\partial P}{\partial z} = g \cos\alpha$ (3)
The free water surface equation:
$B_\eta \frac{\partial \eta}{\partial t} = \frac{\partial}{\partial x}\int_\eta^h UB\,dz - \int_\eta^h qB\,dz$ (4)
The equation of state:
$\rho = f(T_W, \Phi_{TDS}, \Phi_{ISS})$ (5)
Accurate hydrodynamic calculations require accurate water densities. Water
densities are affected by variations in temperature and the concentration of solids. The
following relationship is used in the model:
$\rho_{Tw} = 999.845259 + 6.793952 \times 10^{-2}T_w - 9.19529 \times 10^{-3}T_w^2 + 1.001685 \times$
$10^{-4}T_w^3 - 1.120083 \times 10^{-6}T_w^4 + 6.536332 \times 10^{-9}T_w^5$ (6)
where $x$ and $z$ represent the horizontal distance and vertical elevation, respectively; $U$
and $W$ are the temporal mean velocity components in the horizontal and vertical
directions; $B$ is the channel width; $q$ is the discharge; $t$ denotes the time; $g$ is the


acceleration of gravity; α is the angle of the riverbed with respect to the
x-direction; $P$ represents pressure; $\tau_{xx}$ and $\tau_{xz}$ are the lateral average shear stress
in the $x$-direction and $z$-direction, respectively; $\rho$ and $\rho_{Tw}$ represent densities; $\eta$
and $h$ are the water surface and water depth, respectively; and $T_W$ is the water
temperature.

The universal transport equation for scalar variables, such as temperature and

chemical oxygen demand (COD), is as follows:
$$\frac{\partial B\Phi}{\partial t} + \frac{\partial UB\Phi}{\partial x} + \frac{\partial WB\Phi}{\partial z} - \frac{\partial\left(BD_x\frac{\partial\Phi}{\partial x}\right)}{\partial x} - \frac{\left(BD_z\frac{\partial\Phi}{\partial z}\right)}{-\partial z} = q_\Phi B + S_\Phi B \qquad (7)$$
where $\Phi$ is the laterally averaged constituent concentration; $D_x$ and $D_z$ are the
temperature and constituent dispersion coefficient in the horizontal and vertical
directions, respectively; $q_\Phi$ represents the lateral inflow or outflow mass flow rate of
the constituent per unit volume; and $S_\Phi$ denotes the laterally averaged source/sink
term.

Heat exchange at the water surface includes net solar shortwave radiation, net

longwave radiation, evaporation and conduction. The surface heat exchange is
computed as follows:
$$H_n = H_s + H_a + H_e + H_c - (H_{sr} + H_{ar} + H_{br}) \qquad (8)$$
where $H_n$ is the net rate of heat exchange across the water surface; $H_s$ is the
incident shortwave solar radiation; $H_a$ represents the incident longwave radiation;
$H_{sr}$ and $H_{ar}$ represent the reflected solar radiation of shortwave and longwave
radiation, respectively; $H_{br}$ is the back radiation from the water surface; $H_e$ is the





evaporative heat loss; and $H_c$ represents the heat conduction.
**2.2.2 Model validation**
The water quality at the Tangxi River Bridge was monitored in 2017, and the data
were used to verify the model. The Tangxi River Bridge is 18 km from the confluence.
Due to the low water level of the main reservoir, the backwater did not reach the
Tangxi River Bridge from June to August. Therefore, only the data from January to
May and from September to December were selected to verify the simulated results of
water temperature (T), ammonia nitrogen (NH$_3$-N), total phosphorus (TP), and total
nitrogen (TN). COD values were not measured.
The results showed that the simulated values of T, TP and TN fit well with the
measured values. The minimum difference in T between the simulated value and the
measured value was 0.6 ℃, the maximum difference was 4.7 ℃, and the error
percentage between the simulated values and the measured values ranged from 3 -
29%. The minimum difference in TP between the simulated values and the measured
values was 0.004 mg/L, the maximum difference was 0.03 mg/L, and the error
percentage between the simulated and measured values ranged from 5 - 34%. The
minimum and maximum differences in TN between the simulated and measured
values were 0.02 mg/L and 0.26 mg/L, respectively, and the error percentage ranged
from 3 - 38%. For NH$_3$-N, the differences between the simulated and measured values
were greater than 0.3 mg/L, and the error percentage was greater than 30%. The
degradation process of NH$_3$-N usually exhibits characteristics and there are many






factors affecting the degradation coefficient of NH₃-N, such as the microbial
properties of the water, hydrodynamic conditions, water pollution degree, suspended
solids and pH (Bockelmann et al., 2004; Wang et al., 2016; Pan et al., 2020), which
resulted in a higher simulation error than other values.

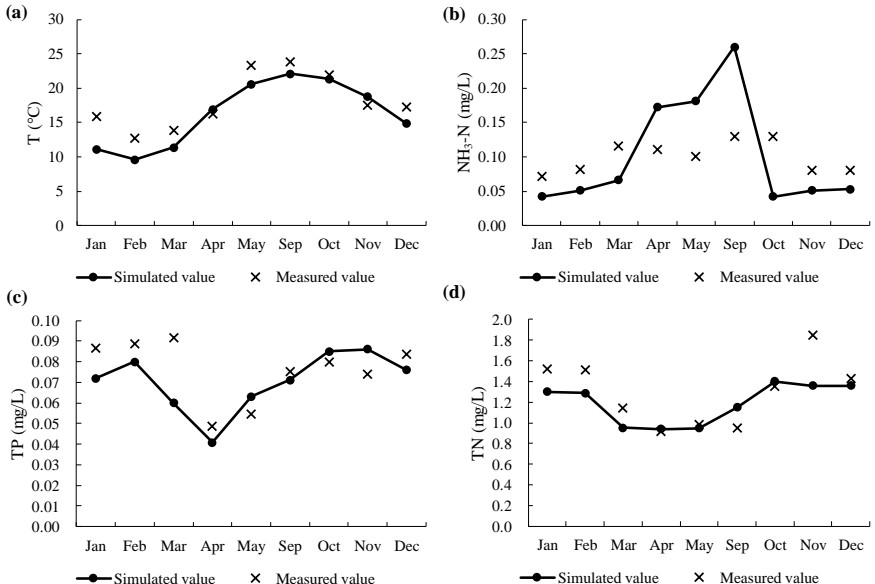


**Fig. 2.** The comparison between the simulated and measured values at the Tangxi
River Bridge in each month. (a) Comparison of water temperature, (b) comparison of
ammonia nitrogen, (c) comparison of total phosphorus, (d) comparison of total
nitrogen. The points on the graph are simulated values, and the cross marks on the
graph are measured values.
**2.2.3 Boundary conditions**
The boundary conditions of the calculation included the meteorology, water
temperature of the inflow, discharge flow, water quality and water level of the TGR.





The meteorological conditions of the Tangxi River and TGR were based on
meteorological data from Yunyang County (Table 1), and the pollution loads of point
and non-point sources were counted and then calculated in this study (Table 2). The
boundary conditions of flow, water level and water quality are shown in Fig. 3.
**Table 1.**
Statistical table of meteorological data from the Yunyang meteorological station.

| Month | Temperature | Wind speed | Wind direction | Cloudiness | Solar radiation | Relative humidity |
|---|---|---|---|---|---|---|
| | °C | m/s | ° | % | W/m$^2$ | % |
| 1 | 7.6 | 0.8 | 146 | 81 | 57.1 | 78.5 |
| 2 | 9.8 | 0.9 | 178 | 82 | 74.3 | 75.8 |
| 3 | 14.3 | 1.0 | 165 | 78 | 121.2 | 72.7 |
| 4 | 19.0 | 1.1 | 196 | 75 | 146.3 | 74.6 |
| 5 | 22.9 | 1.1 | 185 | 77 | 149.1 | 76.9 |
| 6 | 25.8 | 1.1 | 198 | 78 | 158.7 | 78.6 |
| 7 | 29.1 | 1.2 | 189 | 68 | 197.5 | 72.9 |
| 8 | 29.0 | 1.2 | 198 | 60 | 203.9 | 69.4 |
| 9 | 24.7 | 1.1 | 216 | 71 | 138.3 | 76.4 |
| 10 | 19.6 | 0.9 | 171 | 78 | 103.9 | 81.4 |
| 11 | 14.5 | 0.8 | 179 | 77 | 73.0 | 83.0 |
| 12 | 9.1 | 0.8 | 172 | 81 | 55.5 | 82.4 |
| Annual | 18.8 | 1.0 | 183 | 76 | 123.2 | 76.9 |

**Table 2.**
Statistics of pollution load in the Tangxi River research area.

| Factors | COD (t/a) | | NH$_3$-N (t/a) | | TP (t/a) | | TN(t/a) | |
|---|---|---|---|---|---|---|---|---|
| | Point | Non-point | Point | Non-point | Point | Non-point | Point | Non-point |
| Pollution Load | 2093.58 | 1537.35 | 354.21 | 154.46 | 35.08 | 23.90 | 2093.58 | 1537.35 |

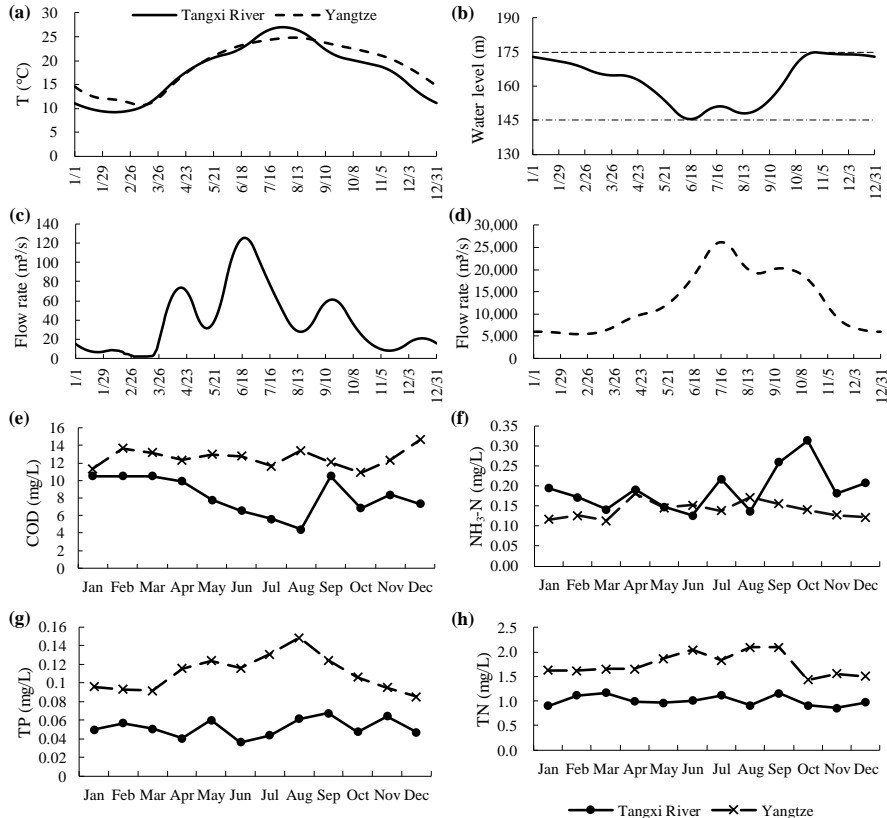


**Fig. 3.** Simulation boundary conditions. (a) Daily water temperatures of the main

reservoir and tail of the tributary bay, (b) water level of the main reservoir, (c) daily

inflow of the tributary bay, (d) daily inflow of the main reservoir, (e) - (h) monthly

water quality (COD, NH₃-N, TP and TN) of the main reservoir and tributary bay,

respectively.

**2.3 Simulation of eutrophication**

The comprehensive nutrition index (TLI($\sum$)) method (Carlson, 1977) was used to

evaluate the nutritional status of the tributary bay. Lakes and reservoirs can be

classified into different nutritional statuses based on their TLI($\sum$) values:





TLI($\sum$)<30, oligotrophic

30≤TLI($\sum$)≤50, mesotrophic

TLI($\sum$)>50, eutrophic

50<TLI($\sum$)≤60, slightly eutrophic

60<TLI($\sum$)≤70, moderately eutrophic

TLI($\sum$) >70, severely eutrophic

The formula for calculating the *TLI( $\Sigma$ )* is as follows:

$TLI(\sum) = \sum_{j=1}^{m} W_j \cdot TLI(j)$                         (9)
where *TLI( $\Sigma$ )* is the comprehensive nutrition index; $W_j$ represents the correlation
weight of the nutrition state index of the *j*-th parameter; and *TLI(j)* denotes the
nutritional status index of the *j*-th parameter.

Considering chlorophyll-a (chla) as the reference parameter, the normalized

correlation weight formula of the *j*-th parameter is as follows:
$W_j = \dfrac{r_{ij}^2}{\sum_{j=1}^{m} r_{ij}^2}$                         (10)
where $r_{ij}$ is the correlation coefficient between the *j*-th parameter and the reference
parameter chla and *m* represents the number of evaluation parameters.

The correlation coefficients $r_{ij}$ and $r_{ij}^2$ between chla and other parameters are

shown in Table 3 (Li and Zhang, 1993).







**Table 3**
The correlation coefficients $r_{ij}$ and $r_{ij}^2$ between chla and other parameters.

| Parameter | TP | TN | SD | COD$_{Mn}$ |
|---|---|---|---|---|
| $r_{ij}$ | 0.84 | 0.82 | -0.83 | 0.83 |
| $r_{ij}^2$ | 0.7056 | 0.6724 | 0.6889 | 0.6889 |

The calculation formula of the nutritional status index of each parameter is shown
as follows:
$$TLI(TP) = 10(9.436 + 1.624 \ln TP) \tag{11}$$
$$TLI(TN) = 10(5.453 + 1.694 \ln TN) \tag{12}$$
$$TLI(SD) = 10(5.118 + 1.94 \ln SD) \tag{13}$$
$$TLI(COD_{Mn}) = 10(0.109 + 2.661 \ln COD_{Mn}) \tag{14}$$
where *TP* is total phosphorus; *TN* represents the total nitrogen; *SD* represents the
Secchi depth, a measure of transparency; and *COD$_{Mn}$* is the chemical oxygen demand.
In the parameters above, TP and TN are pivotal. Limitation of one of these, TP or
TN, can limit algae blooms (Bennett et al., 2017; Morgenstern et al., 2015; Lewis et
al., 2011). The nutrient statuses of the surface water in the Tangxi River tributary bay
in different months were evaluated in this study according to the TLI($\Sigma$) method. The
influence of water temperature was also considered during the nutrient status
evaluation.




## 271  3 Results and discussion

### 272  3.1 Hydrological situation

The temporal variations in confluence flow and water level are shown in Fig. 4a.
During July and from August to October, the flow value at the confluence was
negative, which indicated that the tributary bay was mainly affected by backwater
intrusion from the main reservoir. In contrast, the tributary bay was mainly affected
by the backwater jacking of main reservoir in other months (January - June and
November - December). With the water level fluctuation through the whole year, the
backwater intrusion weakened when the water level of the main reservoir dropped,
and when the water level of the main reservoir rose, the backwater intrusion became
obvious.
The temporal variation in confluence flow and the length of backwater are shown
in Fig. 4b. With the change in the flow at the confluence, the length of backwater also
changed. During January to April and October to December, the water level of the
main reservoir rose to 160 - 175 m, and the backwater reached distances of 39.8 - 42.6
km from the confluence simultaneously. During May to September, the water level of
the main reservoir remained at 145 - 160 m, and the backwater reached distances of
12.6 - 23.8 km from the confluence.
The water level and the length of backwater had a negative correlation with the
confluence flow. When the water level dropped, the value of the confluence flow was
positive, and the length of backwater decreased. The tributary bay was mainly



affected by the jacking of the main reservoir during this period. Conversely, when the
water level rose, the water flow at the confluence was negative, and the length of the
backwater increased. The tributary bay was mainly affected by backwater intrusion at
this time.

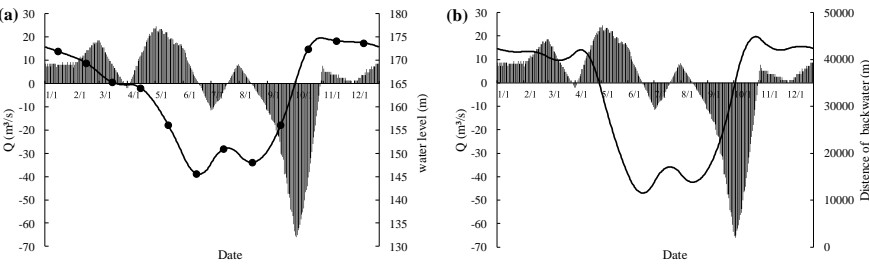

**Fig. 4.** The relationships among water level, length of backwater and confluence flow.
(a) Daily variations in confluence flow and water level and (b) daily variations in
confluence flow and length of backwater.
**3.2 Hydrodynamics**
The distribution of the flow field in each month is shown in Fig. 5. In each month, the
water from the tail flowed along the surface of the tributary bay or sank to the bottom.
The backwater from the main reservoir entered the confluence at different depths
simultaneously, forming one or two flow circulation patterns in the tributary bay.

In response to the jacking of the main reservoir in January, the water from the tail

of the tributary bay first flowed along the surface and then sank to the bottom. Under
the influence of geography, the backwater from the main reservoir formed two large
counterclockwise circulations in the tributary bay. The water level gradually
decreased from February to March, and the backwater effect of the main reservoir





310 also gradually weakened. The water from the tail formed one circulation (February) or

311 two circulations (March) in the tributary bay. From April to June, as the upstream

312 water of the tributary bay joined the surface layer, the circulation zone disappeared.

313 The upstream water gradually sank as it neared the confluence, and at the same time,

314 the backwater from the main reservoir entered the tributary bay in the upper middle

315 layers and formed a small counterclockwise circulation. From July to August, the

316 upstream water of the tributary bay directly flowed to the confluence from the surface

317 layer, and the backwater from the main reservoir entered the tributary bay in the

318 middle and lower layers, forming one circulation in August and two circulations in

319 July. In September, the upstream water first flowed through the surface layer and then

320 sank to the middle of the tributary bay. The backwater from the main reservoir

321 inclined upward from the lower layer and formed two circulations. The upper

322 circulation was a smaller clockwise circulation, while the lower circulation was a

323 larger counterclockwise circulation. The water level increased significantly from

324 October to December, and the influence of the backwater increased simultaneously.

325 The upstream water of the tributary bay flowed through the surface layer and then

326 sank to the bottom.

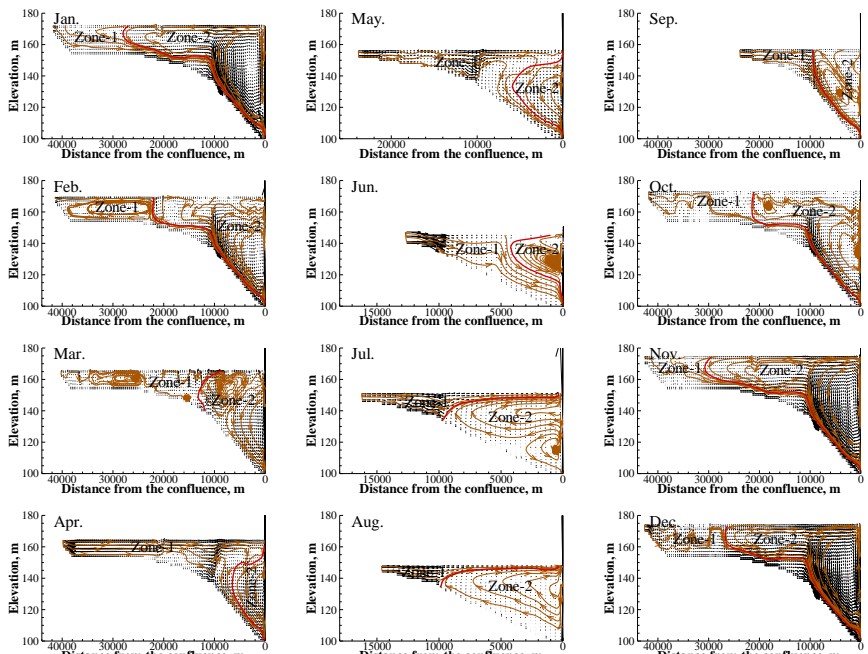

**Fig. 5.** The distribution of the flow field in each month. The flow field was divided

into two areas (Zone 1 and Zone 2) according to the flow field characteristics. The red

curve in the figure is the boundary between Zone 1 and Zone 2.

According to the distribution of the flow field, the tributary bay was divided into

two different areas. Zone 1 represented the area mainly affected by the water from the

tail of the tributary bay, and Zone 2 was the area mainly affected by the backwater

from the main reservoir. Due to the variations in water level and flow value, the

ranges of Zone 1 and Zone 2 differed in each month. The proportions of Zone 1 and

Zone 2 varied with the water level and time (Fig. 6). From January to April, the

backwater reach was from the confluence to Jiangkou Town. With the decrease in the

water levels, the proportion of Zone 1 increased, while the proportion of Zone 2



decreased. From May to September, the length of backwater decreased, and it only
reached Nanxi Town. With the fluctuation in the water level in these months, the trend
of the proportions of Zone 1 and Zone 2 became irregular. From October to November,
with the rise in the water level, the proportion of Zone 1 decreased, while the
proportion of Zone 2 increased. The opposite results were obtained from November to
December when the water level gradually decreased. From October to December, the
backwater again reached Jiangkou Town. These results suggested that the backwater
had a greater impact on the tributary bay when the main reservoir was at a high water
level and had a smaller impact when the main reservoir was at a low water level.

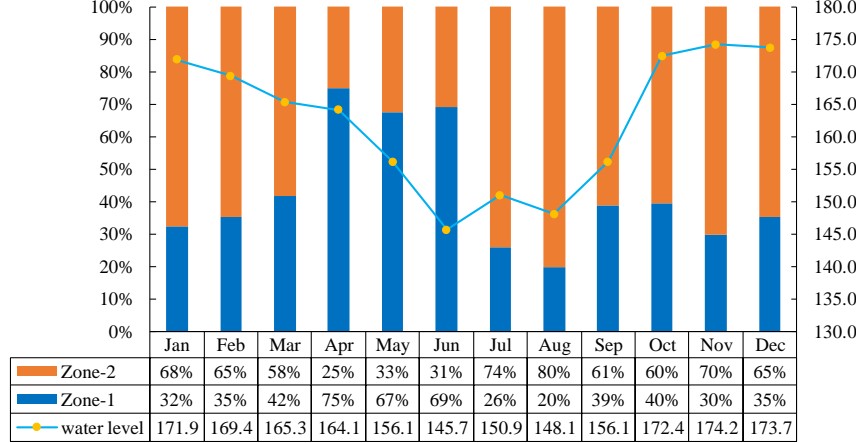


**Fig. 6.** The proportions of Zone 1 and Zone 2 and the variation in water level. The
orange bar represents Zone 2, and the blue bar represents Zone 1. The blue dashed
line represents the variation in water level.
**3.3. Water temperature**
The water temperature distribution of the tributary bay in different months is



shown in Fig. 7. From January to February, July to August, and October to December,
the water temperatures in Zone 1 and Zone 2 were quite different. There was an
obvious temperature boundary, which was mainly affected by the large difference
between the upstream water temperature in the tributary bay and the backwater
temperature from the main reservoir. From March to June and in September, the water
temperature in Zone 1 was similar to that of Zone 2 due to the small difference
between the water temperature at the tail of the tributary bay and the water
temperature of the backwater from the main reservoir.

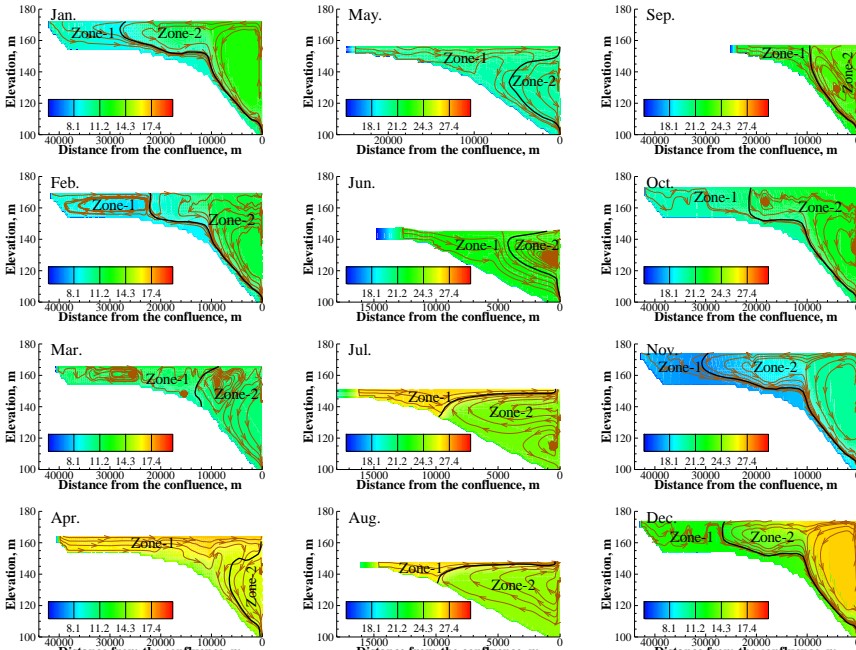


**Fig. 7.** The vertical two-dimensional distribution of water temperature in different
months. The black curve in the figure is the boundary between Zone 1 and Zone 2.

The surface water temperatures of the tributary bay in each month are shown in





Fig. 8a. From March to June, due to the small difference between the upstream water
temperature of the tributary bay and the backwater temperature of the main reservoir,
the surface water temperature changed gently across the bay. The water temperature
gradually decreased from the confluence to the tail of the tributary bay from July to
August and gradually increased from September to October. The water temperature in
the middle reaches was slightly lower than the temperature at the confluence and the
tail of the tributary bay from January to February and from November to December.

The vertical water temperature in the confluence is shown in Fig. 8b. Affected by

solar radiation and air temperature, the water temperature at the surface was relatively
higher than that at the bottom (Zeng et al., 2016; Carey et al., 2012). The temperature
in the middle layers changed little. There was a small thermocline in the surface water
from May to August, and sinking of cold water occurred in January, February, and
September to December.

The average water temperatures of Zone 1 and Zone 2 in different months are

shown in Fig. 8c. The average water temperatures of Zone 1 and Zone 2 were similar
from March to June and in September, while a difference of more than 1.5 ℃ existed
in other months. As the water of Zone 1 mainly came from the upstream of the
tributary bay, it was significantly affected by the air temperature (Mohseni and Stefan,
1999). Zone 2 was mainly affected by the backwater from the main reservoir.
Therefore, the average water temperature in Zone 1 was higher than that in Zone 2 in
summer, and the average water temperature in Zone 1 was lower than that in Zone 2
in winter.

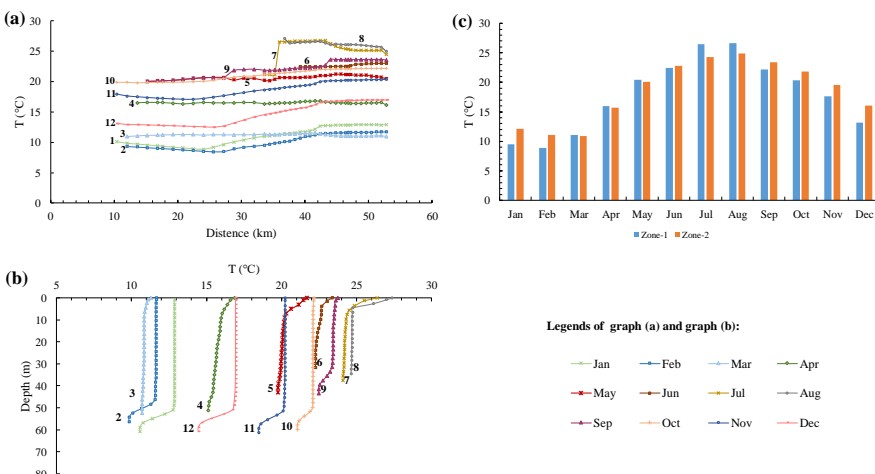

**Fig. 8.** Changes in water temperature. (a) The variation in surface water temperature
in each month along the tributary bay, (b) the variation in the vertical water
temperature at the confluence in each month, and (c) the average water temperatures
of Zone 1 and Zone 2 in each month. The blue bar represents Zone 1, and the orange
bar represents Zone 2 in panel (c).
**3.4 Water quality**
As shown in Fig. 9, the COD concentration in the tributary bay ranged from 0 - 13
mg/L. There was no significant difference in COD concentrations between the tail of
the tributary bay and the backwater from the main reservoir, both of which had values
between 8 and 11 mg/L. With a decreasing trend along the bay, the concentration of
COD reached a minimum value at the intersection of Zone 1 and Zone 2.

The $NH_3$-N concentration in the tributary bay was in the range of 0 - 0.3 mg/L

(Fig. 10). Since the concentration of $NH_3$-N in the tail of the tributary bay was higher



than that of the backwater from the main reservoir, the concentration of $NH_3$-N in
Zone 1 was higher than that in Zone 2 from January to March and July to December.
There was no significant difference in $NH_3$-N between the tail of the tributary bay and
the backwater from the main reservoir in April to June. Additionally, with a
decreasing trend along the bay, the concentration of $NH_3$-N was lower at the
intersection of Zones 1 and 2 than at the tail of the tributary bay or the confluence.
The distributions of TP and TN proved that the nutrients in tributary bays did not
originate solely in the tributary bays but instead were mainly from the main reservoir
and that the nutrient levels were different across seasons. The distributions of TP and
TN in the tributary bay were almost the same. The concentration near the confluence
was relatively high. With the mixing of the water from the tail of the tributary bay and
the backwater from the main reservoir and with the degradation of water quality, the
concentrations of TP and TN gradually decreased. In particular, the concentration of
TP was in the range of 0.04 - 0.12 mg/L, and the concentration of TN was in the range
of 0.8 - 2.1 mg/L. The concentrations of TP and TN in Zone 2 were higher than those
in Zone 1. There was an obvious quality concentration boundary in the tributary bay,
which was basically consistent with the regional boundary of the flow field.
Furthermore, there was an obvious transition zone near the quality boundary in
January to May and September to December, while the transition zone in June to
August was very weak.



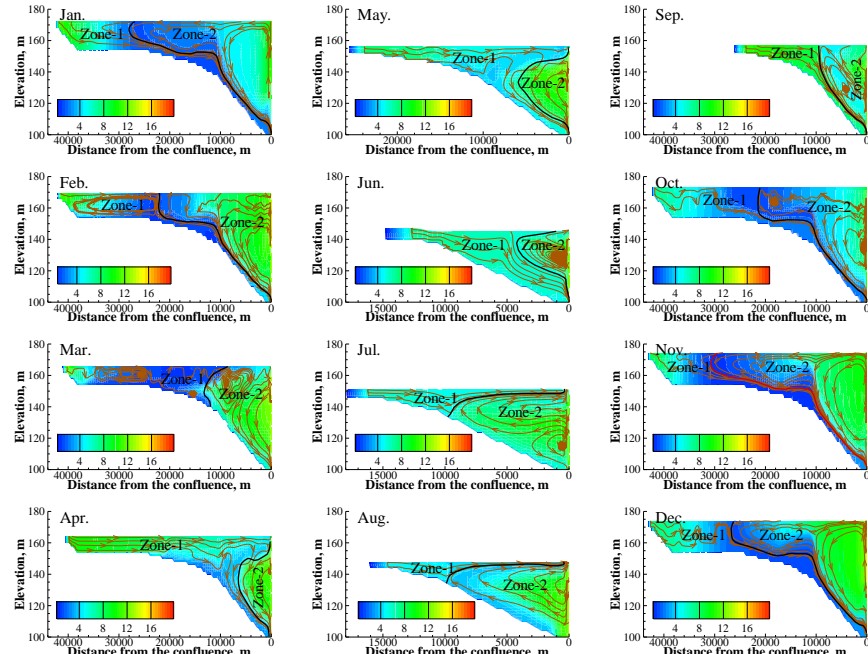

**Fig. 9.** The vertical two-dimensional distribution of COD in each month. The black

curve in the figure is the boundary between Zone 1 and Zone 2.



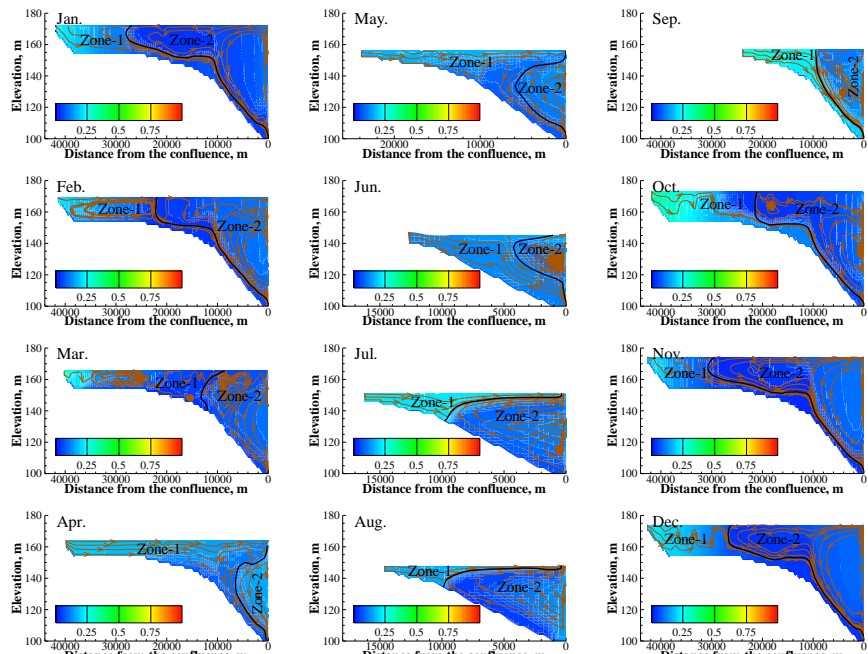

**Fig. 10.** The vertical two-dimensional distribution of NH$_3$-N in each month. The black

curve in the figure is the boundary between Zone 1 and Zone 2.



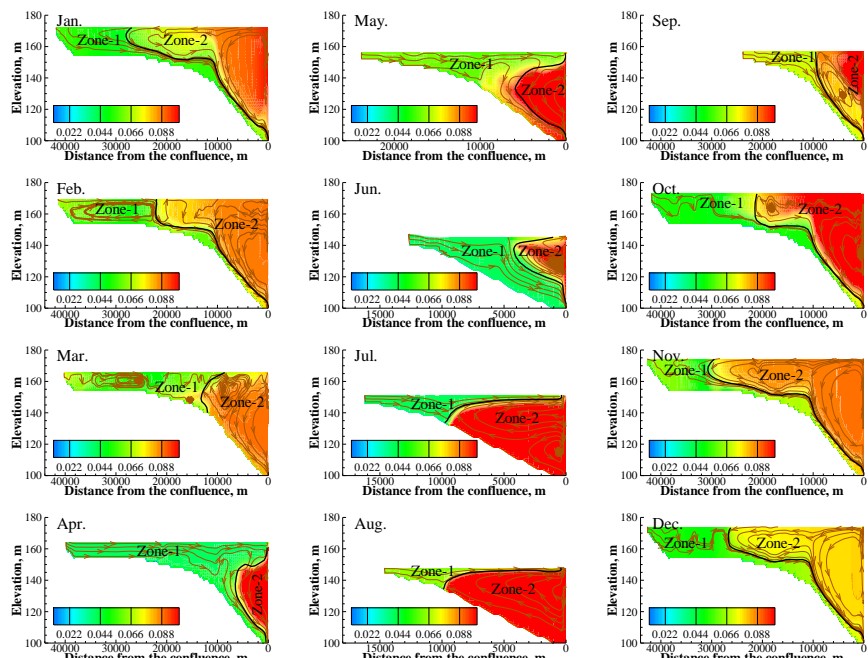

**Fig. 11.** The vertical two-dimensional distribution of TP in each month. The black

curve in the figure is the boundary between Zone 1 and Zone 2.



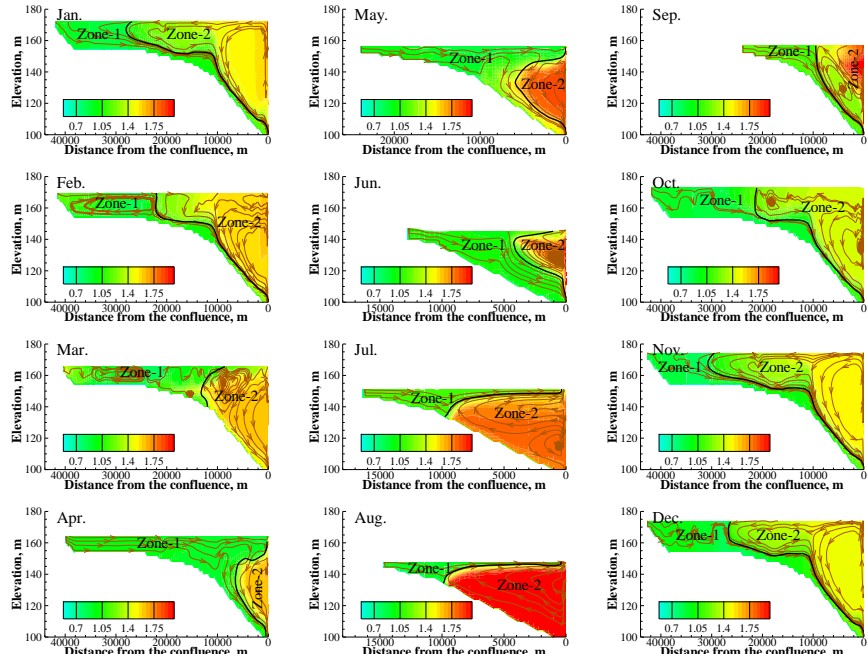

**Fig. 12.** The vertical two-dimensional distribution of TN in each month. The black

curve in the figure is the boundary between Zone 1 and Zone 2.

The COD, NH$_3$-N, TP and TN in the surface water of the tributary bay in different

months are shown in Fig. 13. The concentrations of COD and NH$_3$-N were generally

higher on the two sides and lower in the middle. The concentrations of TP and TN

were higher in the confluence and lower in the tail of the tributary bay.

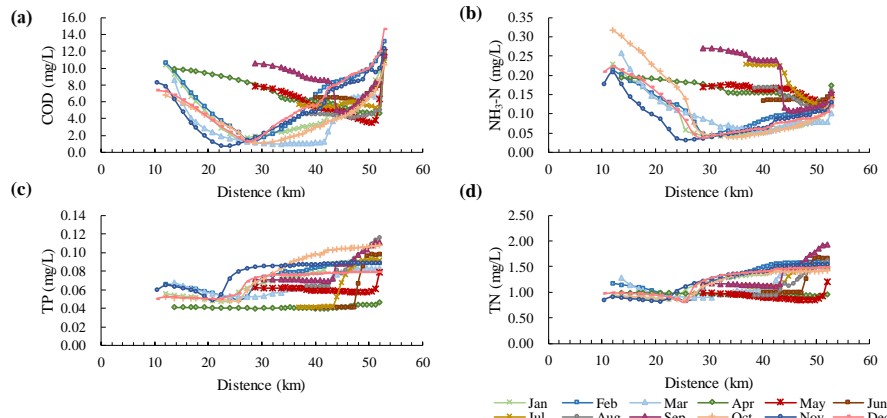

**Fig. 13.** The variation in surface water quality in different months along the tributary

bay. (a) Variation in chemical oxygen demand, (b) variation in ammonia nitrogen, (c)

variation in total phosphorus, and (d) variation in total nitrogen.

The vertical changes in COD, NH$_3$-N, TP and TN in different months at the

confluence are shown in Fig. 14. There was no obvious regularity in the vertical water

quality distributions of COD and NH$_3$-N. The average vertical variation in COD was

4.6 mg/L over 12 months. The largest change appeared in December, with a value of

7.0 mg/L, and the smallest change appeared in June, with a value of 1.6 mg/L. The

average vertical variation in NH$_3$-N was 0.06 mg/L. The largest change appeared in

January, with a value of 0.02 mg/L, and the smallest change appeared in July, with a

value of 0.12 mg/L.

The concentrations of TP and TN were higher in the surface water and lower in

the bottom in January to March and September to December, which was contrary to

that in July and August. From April to June, the concentrations of TP and TN first

increased and then decreased from the surface to the bottom. The concentration
gradient in the upper 10 m surface layer was relatively large.

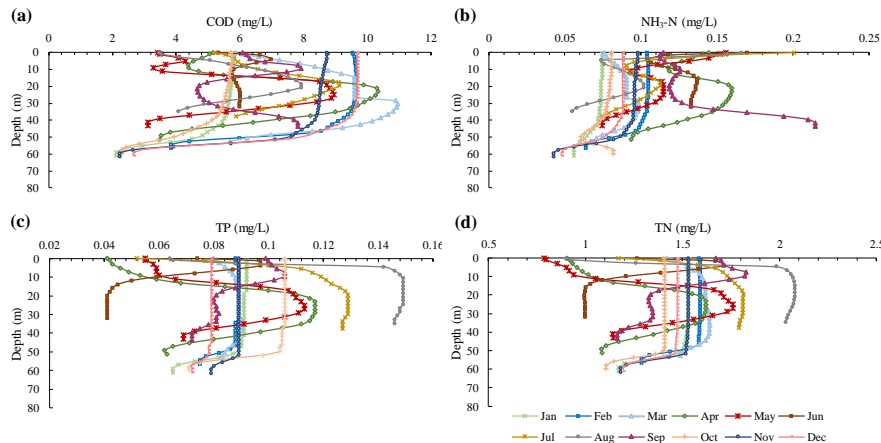


**Fig. 14.** The vertical variation in the water quality in different months at the section
that was 6 km away from the confluence. (a) Variation in chemical oxygen demand, (b)
variation in ammonia nitrogen, (c) variation in total phosphorus, and (d) variation in
total nitrogen.
The average concentrations of COD, $NH_3$-N, TP and TN in Zone 1 and Zone 2 are
shown in Fig. 15. The COD concentration in Zone 2 was higher than that in Zone 1 in
all months except September. The concentration of $NH_3$-N in Zone 1 was higher than
that in Zone 2 due to the higher concentration of $NH_3$-N in the water of the tail of the
tributary bay. For TP and TN, the concentrations in Zone 2 were higher than those in
Zone 1.



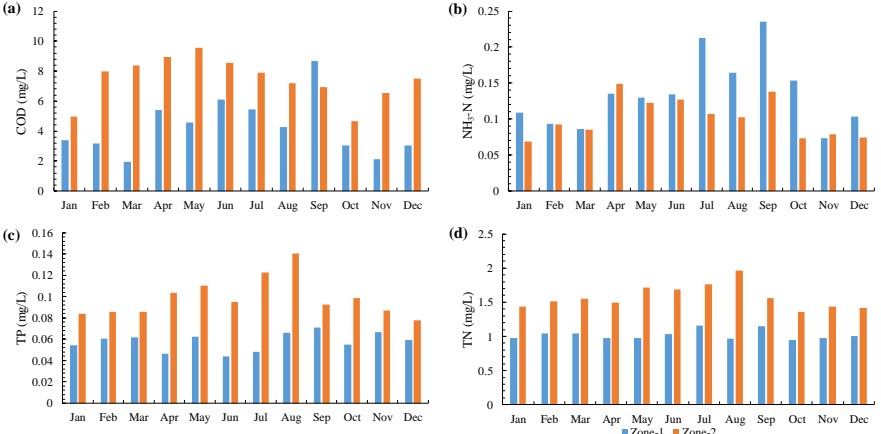


**Fig. 15.** The average water quality changes in Zone 1 and Zone 2. (a) Variation in chemical oxygen demand, (b) variation in ammonia nitrogen, (c) variation in total phosphorus, and (d) variation in total nitrogen. The blue bar represents Zone 1, and the orange bar represents Zone 2.

**3.5 Water eutrophication**

The distribution of the TLI($\sum$) values in the surface water of the tributary bay in different months is shown in Fig. 16. The TLI($\sum$) within 0.5 km of the confluence was relatively higher than in other areas throughout the year, reaching the level of light eutrophication. Additionally, the reach with high TLI($\sum$) values in February and in September to December had a long range. From January to March and September to December, the reach approximately 25 km from the confluence had low TLI($\sum$) values, reaching oligotrophic status. In the rest of the time and area, the TLI($\sum$) values correspond to a medium nutrient level. Additionally, the water temperature near the confluence was less than 20 °C, and the light conditions were poor in January to April





and November to December. Temperature and light conditions are important factors in
the occurrence of eutrophication, and neither low temperatures nor poor light
conditions are conducive to the growth of algae (Singh and Singh, 2015; Romarheim
et al., 2015; Paerl et al., 2011; Reynolds, 2006). Physical dynamics play a critical role
in estuarine biological production, material transport and water quality (Kasai et al.,
2010). The results of this study showed that the tributary bay was mainly affected by
backwater intrusion of the main reservoir in July and from August to October. During
this time, the vertical mixing of water near the confluence was severe, which was also
not conducive to the growth of algae (Gao et al., 2017; Lindim et al., 2011; Huisman
et al., 2006). In conclusion, considering the influence of hydrodynamics, water
temperature and water quality, the risk of eutrophication in the tributary bay was
highest in the section within 0.5 km of the confluence from May to June.

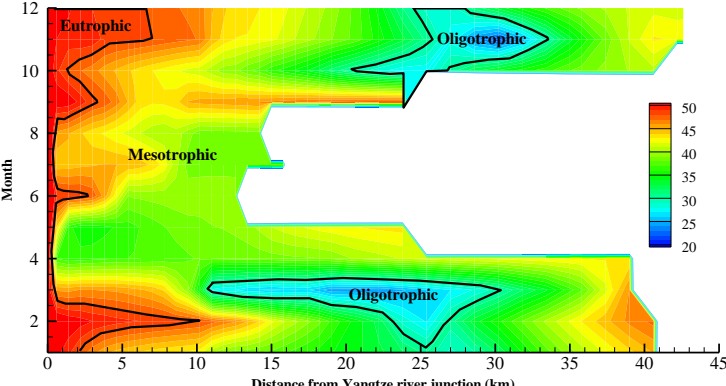


**Fig. 16.** Eutrophication results of surface water in the tributary bay. The nutrient
status of the tributary bay is divided into three states (oligotrophic, mesotrophic and





eutrophic) according to the comprehensive nutrient index.
**4 Conclusions and future work**
In this paper, the effect of the backwater jacking and intrusion of the main reservoir
on the hydrodynamics and water environment of the Tangxi River, a tributary bay of
the TGR are studied. The following conclusions were reached as a result of the
research:
(1) The intrusion was weak when the water level of the main reservoir dropped,
and the tributary bay was mainly affected by the backwater jacking of the main
reservoir. Conversely, when the water level of the main reservoir rose, the tributary
bay was mainly affected by backwater intrusion from the main reservoir. Since the
backwater intrusion brought serve vertical mixing of water that was not conducive to
the growth of algae, the controlling measures of eutrophication could contrapuntally
be proposed in the time that the water level of the main reservoir dropped.
(2) The water from the tail flowed along the surface of the tributary bay or sank to
the bottom in each month. The backwater from the main reservoir entered the
confluence at different depths simultaneously, forming one or two circulations in the
tributary bay. The backwater had a greater impact on the tributary bay when the main
reservoir was at high water level and had a smaller impact when the main reservoir
was at a low water level.
(3) The water temperature of the tributary bay was not greatly affected by the
backwater from the main reservoir. The water qualities in different parts of the



tributary bay were quite different. The concentrations of COD and $NH_3$-N in the
tributary bay were generally higher at the two ends of the bay and lower in the middle.
The concentrations of TP and TN were higher at the confluence and lower at the tail
of the tributary bay. Moreover, for TP and TN, there was an obvious quality
concentration boundary in the tributary bay, which was basically consistent with the
regional boundary of the flow field. The concentrations of TP and TN were higher in
the side near the confluence than that in the other side.
(4) Nutrients in tributary bays were mainly from the main reservoir and the
nutrient levels were affected by constantly changing hydrodynamic conditions and
environmental factors across seasons. According to the simulation of eutrophication,
the TLI($\sum$) values within 0.5 km of the confluence were relatively high. Considering
the influence of hydrodynamics, water temperature and water quality, the risk of
eutrophication of the tributary bay was high within 0.5 km of the confluence in May
and June.
(5) Though the nutrients in tributary bays were mainly from the main reservoir,
the backwater effect of the main reservoir didn't influence the water environment of
the whole tributary bay. Therefore, we can focus on the areas that are more affected
by the main reservoir and propose protective measures targeted at these areas.
This paper only studied the influence of the main reservoir on the tributary bay in
terms of hydrodynamics and water environment. The influence of the tributary bay on
the main reservoir and the interaction between the main reservoir and the tributary



bay are still unclear. In the future, numerical simulation of the main reservoir's
hydrodynamics and water environment based on the results of this paper should be
carried out to explore the interaction between the main reservoir and the tributary bay.

Future work should also explore control measures to improve the water

environment of the tributary bay based on its interaction with the main reservoir. At
present, some scholars have proposed that preventing and controlling eutrophication
in tributary bays can be achieved by the method of "double nutrient reduction", which
involves the simultaneous control of the nutrient inputs from the main stream and the
tributary (Liang et al., 2014). It is also possible to use ecological methods, such as
emergent plants, submerged plants, phytoplankton, benthic organisms and fish, to
improve water eutrophication (Srivastava et al., 2017; Li et al., 2013; Soares et al.,
2011). In addition, the concept of improving the hydrodynamic conditions of the main
stream and controlling the eutrophication of the water body through manually
controlled operation has been widely accepted by many experts and scholars (Yao,
2011; Zheng et al., 2011; Naselli-Flores and Barone, 2005). Based on future research
on the interaction between the main reservoir and the tributary bay with the goal of
ensuring the main function of the main reservoir, water environment protection
measures should be reasonably proposed for tributary bays in the future.
*Declaration of Competing Interest.* We declare that we have no known competing
financial interests or personal relationships that could have appeared to influence the
work reported in this paper.



*Acknowledgements*. This work was sponsored by the fund of Sichuan Province under
permission number 2018SZYZF0001.
*Author contribution*. All co-authors participated in the field collection, data analysis,
and/or writing of this manuscript. Ruifeng Liang was primarily responsible for
preparation and process of this manuscript. Xintong Li and Yuanming Wang
conceived of the study design and data analysis with input from all co-authors.

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
