# Peer review of "Hydrodynamic and environmental characteristics of a tributary bay influenced"

_Hydrology and Earth System Sciences, 2020_

## Referee Comment (RC1) · Anonymous Referee #1 · 27 Feb 2020

The manuscript presents how the backwater jacking and intrusion of the main reservoir influence the hydrodynamic and water environment characteristics of the tributary bay. To my knowledge I, this is likely the first time the main reservoir's backwater jacking and intrusion question is explained clearly. The different effects in different areas of the tributary bay are found. The results can provide guidance for water environment protection in the tributary bays. There are some minor comments listed as below:

1) Line 59 - Line 61: "A tributary bay is always influenced by backwater jacking and intrusion with the rise of the water level of the main reservoir because such changes induce changes in the hydrodynamic conditions in the tributary bay". "the rise of the

[Figure]

water level" is not specific, "fluctuation" is better. And any relevant references for this statement?

2) Introduction section: Please explain what is backwater jacking and what is intrusion, which can make the paper more comprehensible to readers.

3) Line 61- Line 63, Line 64 - Line 66, and Line 91- Line 94: The statements need some more references to support.

4) Line 101- Line 102: Please add the necessity of the study area selection and explain why you select Tangxi River but not other tributaries.

5) Line 220 - Line 221: Please specify the location of the point pollution load.

6) Fig. 4.: It is hard to understand the meaning of fig.4., please add the legend or explain the meaning of the lines in your figure.

7) Line 417 - Line 418: "There was an obvious quality concentration boundary in the tributary bay, which was basically consistent with the regional boundary of the flow field". Are the boundaries of each month in Fig. 9. - Fig. 12. same to the boundaries of each month in Fig. 2. - Fig. 5.? If not, please make a comparison.

8) Fig. 16.: Title of horizontal axis in fig.16. is "... Yangtze River junction", which is not consistent with the previous description "...confluence".

9) What are the degradation coefficients of COD, HN3-N, TP and TN?

---

## Referee Comment (RC2) · Anonymous Referee #2 · 10 Apr 2020

This paper aimed at evaluating the hydrodynamic and water environment effect of backwater jacking and intrusion of the main reservoir on the tributary bay. The topic is novel and of high interest for the relationship between main reservoir and tributary bay. The results are valuable for water environment treatment of the tributary bay. This paper is innovative and suitable to publish in HESS. However, there are also some comments that need to be addressed. After the revision, the paper can be accepted.

Specific comments:

1) Section1 Introduction: Some sentences in Introduction need references to support.

2) Fig.1: The gray area in the upper left picture of Figure 1 should be the area of the

picture in the lower left picture. Some irrelevant places in the upper left picture are marked as gray. Please modify them again.

3) Line 131-139, the reason of selection CE-QUAL-W2 is better to put in introduction part.

4) Section 2 Materials and methods: For the mathematical applications, it is necessary to illustrate the grid division of your study area. It's better to add some explanations or 5) a figure of grid structure in Section 2.

6) Table 1, the format of the temperature unit is messy code. Please correct.

7) TLI($\sum$), $please uniform the format of \sum, in roman or in italics.$

8) Fig. 4, the legend is necessary to be added.

9) Section 2.2.3 Boundary conditions: What was the period ofthe boundary conditions used for simulation?Is it the data of a certain year or the average value of multi-year data? Please specify this in the corresponding section.

10) Section 3.1 Hydrological situation: To my knowledge, density‐driven water can intrude into the tributary bayin the process of TGR impoundment at the end of flood season in autumn, and you specific the backwater intrusion time is from July to October. Do you consider the density‐driven water in your simulation? The intrusion time you specific needs some references to support.

11) Fig. 6: You'd better add titles to the vertical axes to make the figure easier to understand.

12) Section 3.5 Water eutrophication: In your conclusion, the risk of eutrophication in the tributary bay was highest in the section within 0.5 km of the confluence from May to June. Any facts or references in tributary bays of the TGR that can support your conclusion?

13) Line 502- Line 508: You calculated the backwater intrusion time in Section3.1 and

it is a meaningful result. I think you should add this result in the first conclusion.

14) Line 552- Line 555:What is the interaction between the main reservoir and the tributary bay? Asthetributary is a much smaller water body compared with the main stream, so it's easy to understand the influence of main reservoir on tributary.But can the tributary bay affect the main reservoir conversely? I think there needs more details.

15) The conclusion part is better to be condensed and proposed some specific conclusion, or some quantify result.

16) Future work: You mentioned some existing measures to improve the environment of tributary bays, can you propose some possible new methods in your future work section?

---

## Referee Comment (RC3) · Anonymous Referee #3 · 17 Apr 2020

*Review of*
**"The hydrodynamic and environmental characteristics of tributary bay influenced by backwater jacking and intrusion of main reservoir'**
*by X. Li et al.*

This paper reports on an investigation of the effects of water level fluctuations in the Three Gorges Reservoir on a tributary bay on the Tangxi River, the focus being on a number of water quality parameters. The study is based on a numerical simulation using the width-averaged vertically two-dimensional model CE-QUAL-W2. It was conducted for the year 2017 and water quality data collected at the Tangxi River Bridge located 18 km upstream from the confluence was used for validation.

**Major comments**

1. While the results address an important problem they are rather limited in scope. The paper could be enhanced, for example, with a discussion of how sensitive the results are to the model forcing, e.g. winds and air temperature. Are the distributions/variations in the water quality parameters driven solely by the water level fluctuations in the reservoir or do the forcings make a contribution?

2. The model validation is limited to comparisons of water quality parameters at a single point: the Tangxi River Bridge. These measurements do not include measurements of currents so there is no validation of the circulation patterns shown in figure 5 or of the two-dimensional distribution of the water quality patterns. This should be commented on and ideally addressed somehow.

3. The title has some grammatical errors: "The hydrodynamic and environmental characteristics of a tributary bay influenced by backwater jacking and intrusions from a main reservoir'

4. The introduction should include a background discussion on what backwater jacking is and what intrusions from the main reservoir are and the conditions under which they occur. It does not have to be long.

5. The abstract is very long. Seems too long to me.

6. Line 14. " ... is the key ...". Is it really true that this is the one an only key to solving eutrophication or is it one more several. I find it hard to believe that it is the only key to solving these problems. Similarly on line 74. Saying "is a key" seems more accurate.

7. The introduction is very focussed on the Three Gorges Reservoir. The paper could be enhanced by adding a discussion of tributary bays in other parts of the world which would help put the work in a wider context.

8. Line 152. Here it is stated that the water density is affected by concentrations of solids (should be 'suspended solids') but equation (6) for the density is a function of temperature only – it does not depend on concentrations of suspended solids. Were these concentrations included in the model somehow? If so this should be explained. If not this should be made clear.

9. What shortwave absorption model was used in this study? A two- or three-band model, or otherwise? With what attenuation coefficients? Fixed or a function of suspended sediments? In parts of the domain (e.g. figure 5) the water is shallow at some times of the year. Does shortwave radiation reach the bottom? If so how is it handled. Does it reflect off the bottom or is that heat absorbed by the bottom potentially creating unstable stratification?

10. I suggest adding a figure showing some of the meteorological forcings: air temperature and wind in particular. The only information on winds and air temperature are the monthly averages in table 1. Why are averages enough? What was the temporal resolution of the forcings used to drive the model: hourly, daily? Were the monthly averaged values used to driving the model? If so why not more frequent values? No diurnal cycle in the forcing? Is the solar radiation in table 1 a combination of long and short wave radiation? These should be reported separately because shortwave radiation penetration penetrates into the water column and longwave radiation does not.

11. Lines 192–193. The percentage error does not seem like a useful metric. A 25% error for a temperature of $4°$ is very different from a 25% error for a temperature of $20°$.

12. Figure 5. The left side of the region plotted in each panel varies with month of year. How is this left boundary determined? The ranges of $x$ values plotted also varies from month to month which makes it a bit difficult to compare results from different months. The panels are also too small. I find them difficult to read. I suggest full page figures with two columns, all using the same range of $x$ values. Also, the red curve that is the boundary between Zone 1 and Zone 2 is difficult to see because there is not enough contrast with the colours of the other contour lines. They should be very different. In figures 7 and 9 the curve separating the zones is in black. It would be best to use the same colour in all figures. Same comments for other similar figures.

**Minor comments**

1. Line 9. "... by backwater ..." (delete 'the').

2. Line 10. "intrusions from the main reservoir". The main reservoir is not intruding into the bay, it is water from the main reservoir which is intruding.

3. Line 15. "... relevant to the water environment"

4. Line 17. "... by backwater jacking and intrusions from the ..."

5. Line 19. "... and water quality model ..."

6. Line 23. When the water level dropped where? In the main reservoir?

7. Line 24. What is a 'quality concentration boundary'?

8. Line 38. "200 m or even 300 m" is a bit redundant. If dams are 300 m high then it is not necessary to say they are over 200 m high.

9. Line 40. Delete 'However,' and 'the': "These dams block fish .... and change fish communities..."

10. Line 51. "... thus forming water areas ... to lakes known as a tributary bay"

11. Line 90. "... to a rise or decline in chlorophyll content depending ...."

12. Line 91. Do you mean 'Past studies have paid ..."? If you mean the present study (i.e. this paper) then the grammar is incorrect.

13. Line 96. "by backwater jacking and intrusions from the main ..." This needs fixing in many places.

14. Line 96. The sentence "How the .... tributary bay?" needs to be revised. Perhaps "There are many open questions regarding the functions of these types of systems: How does the operation of the main reservoir affect tributary bays?; How do hydrodynamic forces and the water environment of tributary bays respond to backwater jacking and the intrusion of water from the main reservoir?; What controls the water environment of tributary bays?"

15. Line 103. "... by backwater jacking and intrusions from the TGR ..."

16. Line 106. " and water quality ..."

17. Figure 2. The figure caption could be more informative, describing what is shown in each panel.

18. Line 131. "The vertical two-dimensional ...W2 solves the width averaged equations and is appropriate from simulating flow in long narrow water bodies. It was adopted for ..."

19. Line 135. What density current? This is the first mention of a density current.

20. Line 136. "... results using this ..."

21. Line 140. Delete 'listed'.

22. Lines 156–158. This information should appear directly below equations (1)–(5).

23. Line 183. "... was used to ..."

24. Line 200. What does "usually exhibits characteristics" mean? I do not understand this.

25. Line 215. How far away from the tributary bay was the meteorological data collected?

26. Line 216. " sources were calculated and included as inputs to the numerical simulations"

27. Line 265. "... nutrient status of ..."

28. Line 277. Correct grammar.

29. Line 278. Delete "With the water level fluctuation through the whole year"

30. Line 283. "... length of the backwater ..."

31. Line 285. "... main reservoir was between 160 and 175 m and the ..."

32. Figure 4 caption. "The relationships among reservoir water level, length ....". The caption should say what the curves are and what the filled in regions are.

33. Line 302. What is 'water from the tail'?

34. Line 316. What does 'directly flowed to the confluence' mean? Flowed along the surface? This should be clarified. Where is the confluence in the figure?

35. Figure 7. The red contours in the figure should be explained in the caption.

36. Figure 9. Revise caption: "Distribution of COD ...".

37. Line 462. "... was generally higher ..." (it was not higher in every month).

38. Lines 506. I don't understand what the authors are trying to say here: "brought serve vertical"

39. Line 507. What is meant by "could contrapuntally be proposed"?

---

## Author Comment (AC1) · 2 May 2020

**Responses to the referee #1:**

We thank the referee #1 very much for the comments on our manuscript. The comments are valuable during the revision process and will further guide our research. We have studied the comments carefully and revised the manuscript accordingly, which we hope will meet with your approval. The comments (bolded) and responses are fully addressed as follows.

**The manuscript presents how the backwater jacking and intrusion of the main reservoir influence the hydrodynamic and water environment characteristics of the tributary bay. To my knowledge, this is likely the first time the main reservoir's backwater jacking and intrusion question is explained clearly. The different effects in different areas of the tributary bay are found. The results can provide guidance for water environment protection in the tributary bays. There are some minor comments listed as below.**

**Authors' response:** Thank you for your positive and constructive comments. Below we present our responses to each comment.

**1) Line 59 - Line 61: "A tributary bay is always influenced by backwater jacking and intrusion with the rise of the water level of the main reservoir because such changes induce changes in the hydrodynamic conditions in the tributary bay". "the rise of the water level" is not specific, "fluctuation" is better. And any relevant references for this statement?**

**Authors' response:** We have changed "the rise of the water level" to "the fluctuation of the water level" according to your suggestion. We also have added the studies of Ji et al (2010) and Wang et al (2014) as references to support this statement.

**2) Introduction section: Please explain what is backwater jacking and what is intrusion, which can make the paper more comprehensible to readers.**

**Authors' response:** We have added the meanings of backwater jacking and intrusion from the main reservoir in the revised manuscript as follows.

Backwater jacking occurs in tributaries when dams or other obstructions raise the surface of the water upstream from them. Intrusion is the process that the water from the mainstream intrudes into the tributaries.

**3) Line 61- Line 63, Line 64 - Line 66, and Line 91- Line 94: The statements need some more references to support.**

**Authors' response:** Thank you for your suggestion. We have added the studies of Hu et al (2013) and Yin et al (2013) to support the statement of Line 61- Line 63, added the studies of Fu et al (2010), Holbach et al (2013) and Yang et al (2013) and to support the statement of Line 64- Line 66, and added the studies of Zhao (2017) and Long et al (2019) to support the statement of Line 91- Line 94.

**4) Line 101- Line 102: Please add the necessity of the study area selection and explain why you select Tangxi River but not other tributaries.**

**Authors' response:** Tangxi River is a typical tributary bay of the TGR, which is influenced by backwater jacking and intrusion severely in recent years. This phenomenon accelerates the deterioration of water environment of Tangxi River. Thus, the Tangxi River was selected as the focus of this study. This information has been added in our revised manuscript.

**5) Line 220 - Line 221: Please specify the location of the point pollution load.**

**Authors' response:** We have specified the location of the point pollution load on Fig.1 in the revised manuscript. The new Fig.1 is shown as follows.

[Figure]

**6) Fig. 4.: It is hard to understand the meaning of fig.4., please add the legend or explain the meaning of the lines in your figure.**

**Authors' response:** We have added the legend of fig.4 as follows.

[Figure]

**7) Line 417 - Line 418: "There was an obvious quality concentration boundary in the tributary bay, which was basically consistent with the regional boundary of the flow field". Are the boundaries of each month in Fig. 9. - Fig. 12. same to the boundaries of each month in Fig. 2. - Fig. 5.? If not, please make a comparison.**

**Authors' response:** Yes, the boundaries of each month in Fig. 9. - Fig. 12. are the same

to the boundaries of each month in Fig. 2. - Fig. 5. We divided the tributary bay into two areas according to the flow field.

**8) Fig. 16.: Title of horizontal axis in fig.16. is ". . . Yangtze River junction", which is not consistent with the previous description ". . .confluence".**

**Authors' response:** We have changed the title of horizontal axis in Fig.16 from ". . . Yangtze River junction" to ". . .confluence". The revised figure is shown as follows.

[Figure]

**9) What are the degradation coefficients of COD, NH₃-N, TP and TN?**

**Authors' response:** The degradation coefficient of COD is 0.0032 $d^{-1}$, the degradation coefficient of NH$_3$-N is 0.0032 $d^{-1}$, the degradation coefficient of TP is 0.0018 $d^{-1}$, the degradation coefficient of TN is 0.0018 $d^{-1}$. We have added the degradation coefficients in our revised manuscript.

**References**

Fu, B., Wu, B., Lu, Y., Xu, Z., Cao, J., Niu, D., Yang, G., and Zhou, Y.: Three gorges project: efforts and challenges for the environment, Progress in Physical Geography, 34(6), 741-754, https://doi.org/10.1177/0309133310370286, 2010.

Holbach, A., Wang, L., Chen, H., Hu, W., Schleicher, N., Zheng, B., and Norra, S.: Water mass interaction in the confluence zone of the Daning River and the Yangtze River—a driving force for algal growth in the Three Gorges Reservoir, Environmental science and Pollution Research, 20(10), 7027-7037, https://doi.org/10.1007/s11356-012-1373-3, 2013.

Hu, N., Ji, D., Liu, D., Huang, Y., Yin, W., Xiong, C., and Zhang, Y.: Field monitoring and numerical simulating on three-dimensional thermal density currents in the estuary of xiangxi river, Applied Mechanics & Materials, 295-298, 1029-1036, https://doi.org/ 10.4028/www.scientific.net/AMM.295-298.1029, 2013.

Ji, D., Liu, D., Yang, Z. and Xiao, S.: Hydrodynamic characteristics of Xiangxi Bay in Three Gorges Reservoir, Science China (Physics, Mechanics & Astronomy), 40(1), 101-112 (in Chinese), https://doi.org/CNKI:SUN:JGXK.0.2010-01-013, 2010.

Long, L., Ji, D., Yang, Z., Ma, J., Scott, A. W., Liu, D. and Andreas, L.: Density - driven water circulation in a typical tributary of the Three Gorges Reservoir, China, River Research and Application, 35(7), 1-11, https://doi.org/10.1002/rra.3459, 2019.

Wang, Z., Liu, Y., Qin, C., and Zhang, W.: Study on characteristics of hydrodynamic and pollutant transport of the tributary estuary in the three gorges reservoir area, Applied Mechanics & Materials, 675-677, 912-917, https://doi.org/10.4028/www.scientific.net/amm.675-677.912, 2014.

Yang, Z., Liu, D., Ji, D., Xiao, S., Huang, Y. and Ma, J.: An eco-environmental friendly operation: an effective method to mitigate the harmful blooms in the tributary bays of Three Gorges Reservoir, Science China (Technological Sciences), 56, 1458-1470, https://doi.org/10.1007/s11431-013-5190-9, 2013.

Yin, W., Ji, D., Hu, N., Xie, T., Huang, Y., Li, Y. and Zhou, J.: Three-dimensional Water Temperature and Hydrodynamic Simulation of Xiangxi River Estuary, Advanced Materials Research, 726-731(2013), 3212-3221,

https://doi.org/10.4028/www.scientific.net/AMR.726-731.3212, 2013.

Zhao, Y.: Study on the Influence of Mainstream of the Three Gorges Reservoir on Water Quality of Daning River Backwater Area. Ph.D, Tsinghua University, 2017.

---

## Author Comment (AC2) · 2 May 2020

We thank the referee #2 very much for the comments on our manuscript. The comments are valuable during the revision process and will further guide our research. We have studied the comments carefully and revised the manuscript accordingly, which we hope will meet with your approval. The comments (bolded) and responses are fully addressed as follows.

**This paper aimed at evaluating the hydrodynamic and water environment effect of back water jacking and intrusion of the main reservoir on the tributary bay. The topic is novel and of high interest for the relationship between main reservoir and tributary bay. The results are valuable for water environment treatment of the tributary bay. This paper is innovative and suitable to publish in HESS. However, there are also some comments that need to be addressed. After the revision, the paper can be accepted.**

**Authors' response:** Thank you for your positive and constructive comments. Below we present our responses to each comment.

**Specific comments:**

**1) Section 1 Introduction: Some sentences in Introduction need references to support.**

**Authors' response:** Thank you for your suggestion. According to other reviewers' comments, we have added the studies of Ji et al (2010) and Wang et al (2014) to support the statement of Line 59- Line 61, added the studies of Hu et al (2013) and Yin et al (2013) to support the statement of Line 61- Line 63, added the studies of Fu et al (2010), Holbach et al (2013) and Yang et al (2013) to support the statement of Line 64- Line 66, and added the studies of Zhao (2017) and Long et al (2019) to support the statement of Line 91- Line 94.

**2) Fig.1: The gray area in the upper left picture of Figure 1 should be the area of the picture in the lower left picture. Some irrelevant places in the upper left picture are marked as gray. Please modify them again.**

**Authors' response:** We have revised the Fig.1 according to your comment. The new Fig.1 is shown as follows.

[Figure]

**3) Line 131-139, the reason of selection CE-QUAL-W2 is better to put in introduction part.**

**Authors' response:** We have moved the sentences in Line 131 - Line 139 to the last paragraph of the introduction part according to your suggestion.

**4) Section 2 Materials and methods: For the mathematical applications, it is necessary to illustrate the grid division of your study area. It's better to add some explanations.**

**Authors' response:** Thank you for your suggestion. The research river was divided into

107 × 38 (longitudinal × vertical) rectangular cell grids with the longitudinal dimension of 400 m and the vertical dimension of 2 m. The figure of grid structure we added in the revised manuscript is shown in response 5.

**5) A figure of grid structure in Section 2.**

**Authors' response:** We have added the figure of structure in Section 2 in the revised manuscript as follows.

[Figure]

Grid division: 107 × 38 (longitudinal × vertical)

Longitudinal dimension: 400 m

Vertical dimension: 2 m

**6) Table 1, the format of the temperature unit is messy code. Please correct.**

**Authors' response:** We have corrected the format of temperature unit in the revised manuscript.

**7) TLI ($\sum$), please uniform the format of $\sum$, in roman or in italics.**

**Authors' response:** We have uniformed the format of in roman in the revised manuscript.

**8) Fig. 4, the legend is necessary to be added.**

**Authors' response:** We have added the legend of Fig.4 as follows.

[Figure]

**9) Section 2.2.3 Boundary conditions: What was the period of the boundary conditions used for simulation? Is it the data of a certain year or the average value of multi-year data? Please specify this in the corresponding section.**

**Authors' response:** Thank you for your suggestion. The boundary conditions used for simulation were the daily average data of multi-year. This information has been added in Section 2.2.3 in our revised manuscript.

**10) Section 3.1 Hydrological situation: To my knowledge, density driven water can intrude into the tributary bay in the process of TGR impoundment at the end of flood season in autumn, and you specific the backwater intrusion time is from July to October. Do you consider the density driven water in your simulation? The intrusion time you specific needs some references to support.**

**Authors' response:** During the simulation, we considered the influence of density flow, and we have added references to support the intrusion time we specified. The following sentences were added in the revised manuscript.

The periods of intrusion occurred in other tributaries were investigated in previous studies. The backwater intrusion was mainly concentrated in low water level operation period and impoundment period in Daning River (Zhao, 2017). The water of the mainstream of TGR flowed backward into the Xiangxi Bay in density current in different plunging depth during the process of TGR impoundment at the end of flood season in autumn, the intrusion was weak when the water level fell (Ji et al., 2010; Yang

et al., 2018). Compared to the results of previous studies, the backwater intrusion showed obvious seasonal changes and the main intrusion time was almost the same.

**11) Fig. 6: You'd better add titles to the vertical axes to make the figure easier to understand.**

**Authors' response:** Thank you for your suggestion. We have added titles to the vertical axes in Fig.6 and the revised figure is shown as follows.

[Figure]

| | Jan | Feb | Mar | Apr | May | Jun | Jul | Aug | Sep | Oct | Nov | Dec |
|---|---|---|---|---|---|---|---|---|---|---|---|---|
| Zone 2 | 68% | 65% | 58% | 25% | 33% | 31% | 74% | 80% | 61% | 60% | 70% | 65% |
| Zone 1 | 32% | 35% | 42% | 75% | 67% | 69% | 26% | 20% | 39% | 40% | 30% | 35% |
| water level | 171.9 | 169.4 | 165.3 | 164.1 | 156.1 | 145.7 | 150.9 | 148.1 | 156.1 | 172.4 | 174.2 | 173.7 |

**12) Section 3.5 Water eutrophication: In your conclusion, the risk of eutrophication in the tributary bay was highest in the section within 0.5 km of the confluence from May to June. Any facts or references in tributary bays of the TGR that can support your conclusion?**

**Authors' response:** Thank you for your suggestion. We have added the study of Wu et al (2010) to support our conclusion. The following sentences were added in the revised manuscript.

Wu et al (2010) had monitored the eutrophication in Daning River constantly, a tributary bay of the TGR, and found that the algal blooming frequently occurred in the area close to the confluence from March to June, which was similar with present study.

**13) Line 502- Line 508: You calculated the backwater intrusion time in Section 3.1 and it is a meaningful result. I think you should add this result in the first conclusion.**

**Authors' response:** We have added the result of the backwater intrusion time in the first conclusion in the revised manuscript.

**14) Line 552 - Line 555: What is the interaction between the main reservoir and the tributary bay? As the tributary is a much smaller water body compared with the main stream, so it's easy to understand the influence of main reservoir on tributary. But can the tributary bay affect the main reservoir conversely? I think there needs more details.**

**Authors' response:** The interaction between the main reservoir and the tributary bay means their hydrodynamics and water environmental characteristics can influence each other. One tributary bay can affect a small section of main reservoir near its confluence, maybe many tributary bays can influence the main reservoir together. A main reservoir's operation may have common influences on its tributary bays. We have added the following sentences in the revised manuscript.

A main reservoir's operation may have common influences on its tributary bays, and tributary bays may influence the main reservoir together conversely.

**15) The conclusion part is better to be condensed and proposed some specific conclusion, or some quantify result.**

**Authors' response:** Thank you for your suggestion. We have condensed the conclusion part and put the most quantify results in the conclusion part in the revised manuscript.

**16) Future work: You mentioned some existing measures to improve the environment of tributary bays, can you propose some possible new methods in your future work section?**

**Authors' response:** Thank you for your suggestion. We introduced some possible methods in the future work section. At present, the method of "double nutrient reduction", ecological methods, and manually controlled operation method have been proposed by some scholars. We added this information in the future work section in our manuscript.

**References**

Fu, B., Wu, B., Lu, Y., Xu, Z., Cao, J., Niu, D., et al.: Three gorges project: efforts and challenges for the environment, Progress in Physical Geography, 34(6), 741-754, https://doi.org/10.1177/0309133310370286, 2010.

Holbach, A., Wang, L., Chen, H., Hu, W., Schleicher, N., Zheng, B., and Norra, S.: Water mass interaction in the confluence zone of the Daning River and the Yangtze River—a driving force for algal growth in the Three Gorges Reservoir, Environmental science and Pollution Research, 20(10), 7027-7037, https://doi.org/10.1007/s11356-012-1373-3, 2013.

Hu, N., Ji, D., Liu, D., Huang, Y., Yin, W., & Xiong, C., and Zhang, Y.: Field monitoring and numerical simulating on three-dimensional thermal density currents in the estuary of xiangxi river, Applied Mechanics & Materials, 295-298, 1029-1036, https://doi.org/ 10.4028/www.scientific.net/AMM.295-298.1029, 2013.

Ji, D., Liu, D., Yang, Z. and Xiao, S.: Hydrodynamic characteristics of Xiangxi Bay in Three Gorges Reservoir, Science China (Physics, Mechanics & Astronomy), 40(1), 101-112 (in Chinese), https://doi.org/CNKI:SUN:JGXK.0.2010-01-013, 2010.

Ji, D., Liu, D., Yang, Z., and Yu, W.: Adverse slope density flow and its ecological effect on the algae bloom in Xiangxi Bay of TGR during the reservoir impounding at the end of flood season, Journal of Hydraulic Engineering, 41(6), 691-696 (in Chinese), https://doi.org/10.13243/j.cnki.slxb.2010.06.001, 2010.

Long, L., Ji, D., Yang, Z., Ma, J., Scott, A. W., Liu, D. and Andreas, L.: Density - driven water circulation in a typical tributary of the Three Gorges Reservoir, China, River Research and Application, 35(7), 1-11, https://doi.org/10.1002/rra.3459, 2019.

Wang, Z., Liu, Y., Qin, C., and Zhang, W.: Study on characteristics of hydrodynamic and pollutant transport of the tributary estuary in the three gorges reservoir area, Applied Mechanics & Materials, 675-677, 912-917, https://doi.org/10.4028/www.scientific.net/amm.675-677.912, 2014.

Wu, G., Liu, X. and Wan, D.: The Temporal and Spatial Distribution Characteristic of Algare Blooms in the Daninghe River, Environmental Monitoring in China, 26(03), 69-74 (in Chinese), https://doi.org/10.19316/j.issn.1002-6002.2010.03.019, 2010.

Yang, Y., Deng, Y., Xue, W., He, T. and Tuo, Y.: Water temperature and hydrodynamic characteristics of main and branch reservoirs in jinping, Advanced Engineering Sciences, 50(5), 94-101 (in Chinese), https://doi.org/10.15961/j.jsuese.201800054, 2018.

Yang, Z., Liu, D., Ji, D., Xiao, S., Huang, Y. and Ma, J.: An eco-environmental friendly operation: an effective method to mitigate the harmful blooms in the tributary bays of Three Gorges Reservoir, Science China (Technological Sciences), 56, 1458-1470, https://doi.org/10.1007/s11431-013-5190-9, 2013.

Yin, W., Ji, D., Hu, N., Xie, T., Huang, Y., Li, Y. and Zhou, J.: Three-dimensional Water Temperature and Hydrodynamic Simulation of Xiangxi River Estuary, Advanced Materials Research, 726-731(2013), 3212-3221, https://doi.org/10.4028/www.scientific.net/AMR.726-731.3212, 2013.

Zhao, Y.: Study on the Influence of Mainstream of the Three Gorges Reservoir on Water Quality of Daning River Backwater Area. Ph.D, Tsinghua University, 2017.

---

## Author Comment (AC3) · 2 May 2020

**Responses to the referee #3:**

We thank the referee #3 very much for the constructive comments on our manuscript. The comments are valuable during the revision process and will further guide our research. We really appreciate that you give many specific and detailed suggestions, which help enhance our paper a lot. We have studied the comments carefully and revised the manuscript accordingly, which we hope will meet with your approval. The comments (bolded) and responses are fully addressed as follows.

**This paper reports on an investigation of the effects of water level fluctuations in the Three Gorges Reservoir on a tributary bay on the Tangxi River, the focus being on a number of water quality parameters. The study is based on a numerical simulation using the width-averaged vertically two-dimensional model CE-QUAL-W2. It was conducted for the year 2017 and water quality data collected at the Tangxi River Bridge located 18 km upstream from the confluence was used for validation.**

**Authors' response:** Thank you for your commentary. Below we present our responses to each comment.

**Major comments**

**1. While the results address an important problem they are rather limited in scope. The paper could be enhanced, for example, with a discussion of how sensitive the results are to the model forcing, e.g. winds and air temperature. Are the distributions/variations in the water quality parameters driven solely by the water level fluctuations in the reservoir or do the forcings make a contribution?**

**Authors' response:** Thank you for your suggestion. The aim of our paper is to study how a tributary bay was influenced by backwater jacking and intrusion from the main reservoir. The link between the main reservoir and its tributary bay is the hydrodynamic

condition, which is mostly affected by the water level fluctuations (Sha et al., 2015). So, we focused on the water level fluctuations in the main reservoir and its influence on the tributary bay in this manuscript.

We used the daily average data of multi-year on winds and air temperature as the boundary in our simulation. We have discussed the sensitivity of our results to the winds and air temperature at a new section in the revised manuscript.

We also have enhanced our paper in other aspects. For instance, we have added discussions with other tributaries in the results and discussion section. We also have added some references to support our study and added some details to improve the quality of this paper. We hope our efforts to enhance the paper can meet with your approval.

**2. The model validation is limited to comparisons of water quality parameters at a single point: the Tangxi River Bridge. These measurements do not include measurements of currents so there is no validation of the circulation patterns shown in figure 5 or of the two-dimensional distribution of the water quality patterns. This should be commented on and ideally addressed somehow.**

**Authors' response:** Thank you for your comment. We have tried our best to find the fundamental data, but only the data of water quality parameters in Tangxi River Bridge can be got at present. So, we used the data of Tangxi River Bridge to valid the model CE-QUAL-W2. Though the model validation was limited, many scholars have obtained good results by using it. Moreover, this model is mature and has been proved to perform well in simulating the hydrodynamics, water temperature and water quality of reservoirs and lakes. Therefore, we think our results and conclusions are credible. We also have added this information at the end of introduction section. We hope our explanations for this comment can get your understanding and support.

**3. The title has some grammatical errors: "The hydrodynamic and environmental**

**characteristics of a tributary bay influenced by backwater jacking and intrusions from a main reservoir"**

**Authors' response:** We have changed the title according to your suggestion.

**4. The introduction should include a background discussion on what backwater jacking is and what intrusions from the main reservoir are and the conditions under which they occur. It does not have to be long.**

**Authors' response:** We have added the meanings of backwater jacking and intrusion from the main reservoir and the conditions under which they occur in the revised manuscript as follows.

Backwater jacking occurs in tributaries when dams or other obstructions raise the surface of the water upstream from them. Intrusion is the process that the water from the mainstream intrudes into the tributaries.

**5. The abstract is very long. Seems too long to me.**

**Authors' response:** We have condensed the abstract in the revised manuscript.

**6. Line 14. " ... is the key ...". Is it really true that this is the one an only key to solving eutrophication or is it one more several. I find it hard to believe that it is the only key to solving these problems. Similarly on line 74. Saying "is a key" seems more accurate.**

**Authors' response:** We have changed "is the key" to "is a key" in Line 14.

**7. The introduction is very focused on the Three Gorges Reservoir. The paper could be enhanced by adding a discussion of tributary bays in other parts of the world which would help put the work in a wider context.**

**Authors' response:** Thank you for your suggestions. We have tried our best to find the

studies of tributary bays in other parts of the world, but the studies were few. However, we have added discussions of other tributary bays of the TGR in the revised manuscript. We hope our efforts to enhance the paper can meet with your approval.

**8. Line 152. Here it is stated that the water density is affected by concentrations of solids (should be 'suspended solids') but equation (6) for the density is a function of temperature only – it does not depend on concentrations of suspended solids. Were these concentrations included in the model somehow? If so this should be explained. If not this should be made clear.**

**Authors' response:** We are sorry that we made a mistake in this statement. The concentrations of suspended solids weren't included in the model. We have revised this sentence as follows.

Accurate hydrodynamic calculations require accurate water densities. The following equation of state relating the density to the water temperature was used in the model.

**9. What shortwave absorption model was used in this study? A two- or three-band model, or otherwise? With what attenuation coefficients? Fixed or a function of suspended sediments? In parts of the domain (e.g. figure 5) the water is shallow at some times of the year. Does shortwave radiation reach the bottom? If so how is it handled. Does it reflect off the bottom or is that heat absorbed by the bottom potentially creating unstable stratification?**

**Authors' response:** The shortwave absorption we used was according to Bears Law (Thomas and Scott, 2008). The attenuation coefficients in the model include the fraction absorbed at the water surface and the extinction coefficient. The values of them were 0.45 and 0.45 m$^{-1}$ respectively.

As the content of suspended sediments was low in the research area, we didn't consider the suspended sediments in the simulation.

According to our study, the water depth was around 5 m in the upstream from May to September. Most of the shortwave radiation was absorbed by the water, only a small amount of the radiation reached the bottom. Due to the exponential decay of the shortwave radiation, we didn't distinguish the heating after the radiation reached the bottom of the tributary in the simulation.

As for the stratification, the small amount of radiation that reached the bottom of upstream could not cause the vertical convection problem and it had little effect on the stratification. We hope our explanations for this comment can meet with your approval.

**10. I suggest adding a figure showing some of the meteorological forcings: air temperature and wind in particular. The only information on winds and air temperature are the monthly averages in table 1. Why are averages enough? What was the temporal resolution of the forcings used to drive the model: hourly, daily? Were the monthly averaged values used to driving the model? If so why not more frequent values? No diurnal cycle in the forcing? Is the solar radiation in table 1 a combination of long and short wave radiation? These should be reported separately because shortwave radiation penetration penetrates into the water column and longwave radiation does not.**

**Authors' response:** Thank you for your suggestion. Although the meteorological conditions were displayed in the form of monthly average value in the table 1, we used daily average data of multi-year in our simulation. We are sorry that this made you confused.

The diurnal cycle of our simulation last three years. We have added this information in our revised manuscript.

The solar radiation in table 1 was short wave solar radiation and we have specified this in the revised manuscript. The long wave atmospheric radiation was computed from air temperature and cloudiness.

According to your suggestion, we have replaced the table 1 with a figure of daily average values of meteorological data as follows.

[Figure]

**11. Lines 192–193. The percentage error does not seem like a useful metric. A 25% error for a temperature of 4∘ is very different from a 25% error for a temperature of 20∘.**

**Authors' response:** We agree with that the percentage error is not a useful metric, and we have used root mean squared error to reevaluate the model calculation accuracy. We have revised the description of the fitness between simulated values and measured values as follows.

The difference in T between the simulated value and the measured value was 0.6 - 4.7 ºC, and root mean squared error was 1.8 ºC. The difference in TP between the simulated value and the measured value was 0.004 - 0.03 mg/L, and root mean squared error was 0.01 mg/L. The difference in TN between the simulated value and the measured value was 0.02 - 0.26 mg/L, and root mean squared error was 0.16 mg/L. For $NH_3$-N, the difference between the simulated value and the measured value was 0.03 - 0.08 mg/L, root mean squared error was 0.06 mg/L, and the relative error was greater than 30%.

**12. Figure 5. The left side of the region plotted in each panel varies with month of year. How is this left boundary determined? The ranges of *x* values plotted also varies from month to month which makes it a bit difficult to compare results from different months. The panels are also too small. I find them difficult to read. I**

**suggest full page figures with two columns, all using the same range of *x* values. Also, the red curve that is the boundary between Zone 1 and Zone 2 is difficult to see because there is not enough contrast with the colors of the other contour lines. They should be very different. In figures 7 and 9 the curve separating the zones is in black. It would be best to use the same color in all figures. Same comments for other similar figures.**

**Authors' response:** Thank you for your suggestions.

The left boundary was determined by the water depth. We set the minimum number of activation layers in the simulation, and the corresponding water depth is 4 m. The simulation stopped when the water depth is less than 4 m, and the left boundary was determined.

If we put the figures into two columns, the figures will become too long and look not good. So, we still arranged the figures into three columns, and we also ensured the accuracy of the figures. We have output clearer figures in the revised manuscript. We have used the same range of x values and uniformed the color of boundary between Zone 1 and Zone 2 in black in the revised manuscript according to your suggestion. We hope our revisions for these figures can meet with your approval.

**Minor comments**

**1. Line 9. "... by backwater ..." (delete 'the').**

**Authors' response:** We have deleted 'the' in this sentence.

**2. Line 10. "intrusions from the main reservoir". The main reservoir is not intruding into the bay, it is water from the main reservoir which is intruding.**

**Authors' response:** We have changed 'of' to 'from' in this sentence.

**3. Line 15. "... relevant to the water environment"**

**Authors' response:** We have added 'the' in front of 'water environment'.

**4. Line 17. "... by backwater jacking and intrusions from the ..."**

**Authors' response:** We have changed 'of' to 'from' in this sentence.

**5. Line 19. "... and water quality model ..."**

**Authors' response:** We have added 'water' in front of 'quality model'.

**6. Line 23. When the water level dropped where? In the main reservoir?**

**Authors' response:** Yes, in the main reservoir. We have revised this sentence as follows.

The tributary bay was mainly affected by backwater jacking from the main reservoir when the water level of the main reservoir dropped and by intrusion from the main reservoir when the water level of the main reservoir rose.

**7. Line 24. What is a 'quality concentration boundary'?**

**Authors' response:** It is a boundary of the water quality and we have added 'water' in front of 'quality concentration boundary'.

**8. Line 38. "200 m or even 300 m" is a bit redundant. If dams are 300 m high then it is not necessary to say they are over 200 m high.**

**Authors' response:** We have deleted 'over 200 m' in this sentence.

**9. Line 40. Delete 'However,' and 'the': "These dams block fish .... and change fish communities..."**

**Authors' response:** We have deleted them.

**10. Line 51. "... thus forming water areas ... to lakes known as a tributary bay"**

**Authors' response:** We have revised this sentence as follows.

Backwater extends to some tributaries after the construction of dammed-river reservoirs, which causes the water depth to increase and the water velocity to slow in these tributaries, thus forming the water areas similar to lakes known as a tributary bay.

**11. Line 90. "... to a rise or decline in chlorophyll content depending ...."**

**Authors' response:** We have revised this sentence as follows.

Some scholars found that a rise in the water level may lead either to a rise or decline in the chlorophyll content, depending on the water cycle mode in the tributary.

**12. Line 91. Do you mean 'Past studies have paid ..."? If you mean the present study (i.e. this paper) then the grammar is incorrect.**

**Authors' response:** Yes, we mean the past studies. We have changed 'present' to 'past'.

**13. Line 96. "by backwater jacking and intrusions from the main ..." This needs fixing in many places.**

**Authors' response:** We have fixed the mistakes in the revised manuscript about this sentence.

**14. Line 96. The sentence "How the .... tributary bay?" needs to be revised. Perhaps "There are many open questions regarding the functions of these types of systems: How does the operation of the main reservoir affect tributary bays?; How do hydrodynamic forces and the water environment of tributary bays respond to backwater jacking and the intrusion of water from the main reservoir?; What controls the water environment of tributary bays?"**

**Authors' response:** Thank you for your positive and constructive suggestions. We have revised the sentences according to your suggestion.

**15. Line 103. "... by backwater jacking and intrusions from the TGR ..."**

**Authors' response:** We have changed 'of' to 'from' in this sentence.

**16. Line 106. " and water quality ..."**

**Authors' response:** We have added 'water' in front of 'quality model'.

**17. Figure 2. The figure caption could be more informative, describing what is shown in each panel.**

**Authors' response:** We have described each panel in the caption of Figure 2. We also have added descriptions in each panel of Figure 1. The new caption of Figure 1 is as follows.

**Fig. 1.** Research area and hydrologic system of the Tangxi River Basin. (a) The location of research area relative to China; (b) The location of research area relative to Chongqing; (c) Hydrologic system of research area.

**18. Line 131. "The vertical two-dimensional ...W2 solves the width averaged equations and is appropriate from simulating flow in long narrow water bodies. It was adopted for ..."**

**Authors' response:** We have revised this sentence as follows.

The vertical two-dimensional model CE-QUAL-W2 solves the width averaged equations and is appropriate for simulating flow in long narrow water bodies. It was adopted for the calculation of the hydrodynamic conditions, water temperature and water quality in the tributary bay.

**19. Line 135. What density current? This is the first mention of a density current.**

**Authors' response:** It's the density-driven current. We mentioned this to explain the

model can perform well in backwater intrusion issue.

**20. Line 136. "... results using this ..."**

**Authors' response:** We have deleted 'by' in this sentence.

**21. Line 140. Delete 'listed'.**

**Authors' response:** We have deleted 'listed'.

**22. Lines 156–158. This information should appear directly below equations (1) - (5).**

**Authors' response:** We have moved the explanations of each variable below equations (1) - (5).

**23. Line 183. "... was used to ..."**

**Authors' response:** We have corrected this sentence.

**24. Line 200. What does "usually exhibits characteristics" mean? I do not understand this.**

**Authors' response:** We are sorry that we missed a word 'complex'. The correct sentence is "…usually exhibits complex characteristics" and we have corrected this in the revised manuscript.

**25. Line 215. How far away from the tributary bay was the meteorological data collected?**

**Authors' response:** The weather station is about 19.7 km away from the tributary bay. We have added this information in the revised manuscript.

**26. Line 216. " sources were calculated and included as inputs to the numerical**

**simulations"**

**Authors' response:** We have revised this sentence as follows.

The meteorological conditions (Figure 4) of the Tangxi River and TGR were based on the data from Yunyang County weather station (19.7 km away from the tributary bay), and the pollution loads of point and non-point sources were calculated and included as inputs to the numerical simulations (Table 1).

**27. Line 265. "... nutrient status of ..."**

**Authors' response:** We have corrected this sentence.

**28. Line 277. Correct grammar.**

**Authors' response:** We have changed 'of' to 'from' in this sentence.

**29. Line 278. Delete "With the water level fluctuation through the whole year"**

**Authors' response:** We have deleted this.

**30. Line 283. "... length of the backwater ..."**

**Authors' response:** We have added 'the' in front of 'backwater' in this sentence.

**31. Line 285. "... main reservoir was between 160 and 175 m and the ..."**

**Authors' response:** We have revised this sentence as follows.

During January to April and October to December, the water level of the main reservoir between 160 and 175 m and the backwater reached distances of 39.8 - 42.6 km from the confluence simultaneously.

**32. Figure 4 caption. "The relationships among reservoir water level, length ....".**

**The caption should say what the curves are and what the filled in regions are.**

**Authors' response:** Thank you for your suggestion. We have added the legend of fig.4 in the revised manuscript as follows.

[Figure]

**33. Line 302. What is 'water from the tail'?**

**Authors' response:** We have revised this sentence as follows.

In each month, the upstream water flowed along the surface of the tributary bay or sank to the bottom.

**34. Line 316. What does 'directly flowed to the confluence' mean? Flowed along the surface? This should be clarified. Where is the confluence in the figure?**

**Authors' response:** Yes, we meant the upstream water flowed along the surface. The confluence is the right end of the tributary bay in the figure. We have revised this sentence as follows.

From July to August, the upstream water of the tributary bay directly flowed to the confluence along the surface layer.

**35. Figure 7. The red contours in the figure should be explained in the caption.**

**Authors' response:** We have added the explanation of the red contours in the caption. The revised caption of Figure 7 is shown as follows.

**Fig. 7.** The vertical two-dimensional distribution of water temperature in different months. The black curve in the figure is the boundary between Zone 1 and Zone 2. The brown curves with arrows are streamlines.

**36. Figure 9. Revise caption: "Distribution of COD ...".**

**Authors' response:** We have revised the caption of Figure 9 and we also have revised the same question in Fig.10 - Fig.12.

**37. Line 462. "... was generally higher ..." (it was not higher in every month).**

**Authors' response:** We have added 'generally' in front of 'higher' in this sentence.

**38. Lines 506. I don't understand what the authors are trying to say here: "brought serve vertical"**

**Authors' response:** We are sorry that this sentence made you confused, and we have deleted this sentence in the revised manuscript.

**39. Line 507. What is meant by "could contrapuntally be proposed"?**

**Authors' response:** We are sorry we used an inappropriate word 'brought' and we have deleted it in the revised manuscript.

**References**

Sha, Y., Wei, Y., Li, W., Fan, J. and Cheng, C.: Artificial tide generation and its effects on the water environment in the backwater of Three Gorges Reservoir, Journal of Hydrology, 528, 230-237, https://doi.org/10.1016/j.jhydrol.2015.06.020, 2015.

Thomas, M. C. and Scott A. W.: CE-QUAL-W2: A two-dimensional laterally averaged hydrodynamic and water quality model, Version 3.6, Department of Civil and Environmental Engineering, Portland State University, Portland, 2008.

---

## Author Response (AR1)

**Dear Editor and Referees,**

On behalf of my co-authors, I would like to thank you for the insightful comments and for the opportunity to revise our manuscript entitled "Hydrodynamic and environmental characteristics of a tributary bay influenced by backwater jacking and intrusions from a main reservoir" (ID: hess-2020-63).

The comments were valuable during the revision process and will further guide our research. We have carefully addressed the comments with point-by-point replies to the referees and revised the manuscript accordingly, which we hope will meet with your approval. We then sent our revised manuscript to a professional English editing service (American Journal Experts) to further improve the English language prior to resubmission. American Journal Experts has revised our manuscript for proper English language, grammar, punctuation, spelling and overall style, and we have ensured that the intended meaning has been maintained.

In the section below, we have provided detailed responses to the referees' comments and illustrated the primary corrections made in the paper.

**Responses to the referee #1:**

**The manuscript presents how the backwater jacking and intrusion of the main reservoir influence the hydrodynamic and water environment characteristics of the tributary bay. To my knowledge, this is likely the first time the main reservoir's backwater jacking and intrusion question is explained clearly. The different effects in different areas of the tributary bay are found. The results can provide guidance for water environment protection in the tributary bays. There are some minor comments listed as below.**

**Authors' response:** Thank you for your positive and constructive comments. Below we present our responses to each comment.

**1) Line 59 - Line 61: "A tributary bay is always influenced by backwater jacking and intrusion with the rise of the water level of the main reservoir because such changes induce changes in the hydrodynamic conditions in the tributary bay". "the rise of the water level" is not specific, "fluctuation" is better. And any relevant references for this statement?**

**Authors' response:** We have changed "the rise of the water level" to "fluctuations of the water level" according to your suggestion (Page 3, Line 56). We also have added the studies of Ji et al (2010) and Wang et al (2014) as references to support this statement (Page 3, Lines 56 to Line 58).

**2) Introduction section: Please explain what is backwater jacking and what is intrusion, which can make the paper more comprehensible to readers.**

**Authors' response:** We have added the meanings of backwater jacking and intrusions from the main reservoir in the revised manuscript (Page 3, Line 52 to Line 55) as follows.

Backwater jacking occurs in tributaries when dams or other obstructions raise the surface of the water upstream from them. Intrusion is the process by which water from the mainstream intrudes into the tributary.

**3) Line 61- Line 63, Line 64 - Line 66, and Line 91- Line 94: The statements need some more references to support.**

**Authors' response:** Thank you for your suggestion. We have added the studies of Hu et al (2013) and Yin et al (2013) to support the statement at previous Lines 61 - 63, added the studies of Fu et al (2010), Holbach et al (2013) and Yang et al (2013) and to support the statement at previous Lines 64 - 66, and added the studies of Zhao (2017) and Long et al (2019) to support the statement at previous Lines 91 - 94.

**4) Line 101- Line 102: Please add the necessity of the study area selection and explain why you select Tangxi River but not other tributaries.**

**Authors' response:** The Tangxi River is a typical tributary bay of the TGR, and it has been severely influenced by backwater jacking and intrusions in recent years. This phenomenon accelerates the deterioration of the water environment of Tangxi River. Thus, the Tangxi River was selected as the focus of this study. This information has been added in our revised manuscript (Page 5, Line 100 to Line 103).

**5) Line 220 - Line 221: Please specify the location of the point pollution load.**

**Authors' response:** We have specified the location of the point pollution load on Fig.1 in the revised manuscript (Page 7, Fig.1). The new Fig.1 is shown as follows.

[Figure]

**6) Fig. 4.: It is hard to understand the meaning of fig.4., please add the legend or explain the meaning of the lines in your figure.**

**Authors' response:** We have added the legend of this figure as follows (Page 18, Fig.6).

[Figure]

**7) Line 417 - Line 418: "There was an obvious quality concentration boundary in the tributary bay, which was basically consistent with the regional boundary of the flow field". Are the boundaries of each month in Fig. 9. - Fig. 12. same to the boundaries of each month in Fig. 2. - Fig. 5.? If not, please make a comparison.**

**Authors' response:** Yes, the boundaries of each month in previous Fig. 9. - Fig. 12.

are the same to the boundaries of each month in previous Fig. 2. - Fig. 5. We divided the tributary bay into two areas according to the flow field.

**8) Fig. 16.: Title of horizontal axis in fig.16. is ". . . Yangtze River junction", which is not consistent with the previous description ". . .confluence".**

**Authors' response:** We have changed the title of horizontal axis in this figure from ". . . Yangtze River junction" to ". . .confluence". The revised figure is shown as follows (Page 35, Fig.18).

[Figure]

**9) What are the degradation coefficients of COD, NH$_3$-N, TP and TN?**

**Authors' response:** The degradation coefficient of COD is 0.0032 d$^{-1}$, the degradation coefficient of NH$_3$-N is 0.0032 d$^{-1}$, the degradation coefficient of TP is 0.0018 d$^{-1}$, the degradation coefficient of TN is 0.0018 d$^{-1}$. We have added the degradation coefficients in our revised manuscript (Page 11, Line 205 to Line 206).

**Responses to the referee #2:**

**This paper aimed at evaluating the hydrodynamic and water environment effect of back water jacking and intrusion of the main reservoir on the tributary bay. The topic is novel and of high interest for the relationship between main reservoir and tributary bay. The results are valuable for water environment treatment of the tributary bay. This paper is innovative and suitable to publish in HESS. However, there are also some comments that need to be addressed. After the revision, the paper can be accepted.**

**Authors' response:** Thank you for your positive and constructive comments. Below we present our responses to each comment.

**Specific comments:**

**1) Section 1 Introduction: Some sentences in Introduction need references to support.**

**Authors' response:** Thank you for your suggestion. According to other reviewers' comments, we have added the studies of Ji et al (2010) and Wang et al (2014) to support the statement at Lines 55 - 58, added the studies of Hu et al (2013) and Yin et al (2013) to support the statement at Lines 58 - 60, added the studies of Fu et al (2010), Holbach et al (2013) and Yang et al (2013) to support the statement at Lines 61 - 64, and added the studies of Zhao (2017) and Long et al (2019) to support the statement at Lines 89 - 92.

**2) Fig.1: The gray area in the upper left picture of Figure 1 should be the area of the picture in the lower left picture. Some irrelevant places in the upper left picture are marked as gray. Please modify them again.**

**Authors' response:** We have revised the Fig.1 according to your comment in the revised manuscript (Page 7, Fig.1).

**3) Line 131-139, the reason of selection CE-QUAL-W2 is better to put in introduction part.**

**Authors' response:** We have moved the sentences in Line 131 - Line 139 to the last paragraph of the introduction part according to your suggestion (Page 6, Line 106 to Line 111).

**4) Section 2 Materials and methods: For the mathematical applications, it is necessary to illustrate the grid division of your study area. It's better to add some explanations.**

**Authors' response:** Thank you for your suggestion. The research river was divided into 107 × 38 (longitudinal × vertical) rectangular cell grids with the longitudinal dimension of 400 m and the vertical dimension of 2 m. The figure of grid structure we added in the revised manuscript is shown in response 5.

**5) A figure of grid structure in Section 2.**

**Authors' response:** We have added the figure of structure in Section 2 in the revised manuscript as follows (Page 8, Fig.2).

[Figure]

Grid division: 107 × 38 (longitudinal × vertical)
Longitudinal dimension: 400 m
Vertical dimension: 2 m

**6) Table 1, the format of the temperature unit is messy code. Please correct.**

**Authors' response:** We have corrected the format of temperature unit and replaced the table 1 with a figure in the revised manuscript (Page 13, Fig.4).

**7) TLI ($\sum$), please uniform the format of $\sum$, in roman or in italics.**

**Authors' response:** We have uniformed the format of in roman in the revised manuscript.

**8) Fig. 4, the legend is necessary to be added.**

**Authors' response:** We have added the legend of this figure in the revised manuscript (Page 18, Fig.6).

**9) Section 2.2.3 Boundary conditions: What was the period of the boundary conditions used for simulation? Is it the data of a certain year or the average value of multi-year data? Please specify this in the corresponding section.**

**Authors' response:** Thank you for your suggestion. The boundary conditions used for simulation were the daily average data of multi-year. This information has been added in Section 2.2.3 in our revised manuscript (Page 13, Line 232 to Line 234).

**10) Section 3.1 Hydrological situation: To my knowledge, density driven water can intrude into the tributary bay in the process of TGR impoundment at the end of flood season in autumn, and you specific the backwater intrusion time is from July to October. Do you consider the density driven water in your simulation? The intrusion time you specific needs some references to support.**

**Authors' response:** During the simulation, we considered the influence of density flow, and we have added references to support the intrusion time we specified. The following sentences were added in the revised manuscript (Page 17, Line 295 to Line 303).

Periods of intrusions that occurred in other tributaries were investigated in previous studies. Backwater intrusions were mainly concentrated in low water level operation period and impoundment period in the Daning River (Zhao, 2017). The water of the mainstream of TGR flowed backward into the Xiangxi Bay in the density current at different plunging depths during the process of TGR impoundment at the end of the flood season in autumn, and the intrusion was weak when the water level fell (Ji et al., 2010; Yang et al., 2018). Compared to the results of previous studies, the backwater intrusions showed obvious seasonal changes and the main intrusion time was almost the same.

**11) Fig. 6: You'd better add titles to the vertical axes to make the figure easier to understand.**

**Authors' response:** Thank you for your suggestion. We have added titles to the vertical axes in this figure and the revised figure is shown as follows (Page 21, Fig.8).

[Figure]

| | Jan | Feb | Mar | Apr | May | Jun | Jul | Aug | Sep | Oct | Nov | Dec |
|---|---|---|---|---|---|---|---|---|---|---|---|---|
| Zone 2 | 68% | 65% | 58% | 25% | 33% | 31% | 74% | 80% | 61% | 60% | 70% | 65% |
| Zone 1 | 32% | 35% | 42% | 75% | 67% | 69% | 26% | 20% | 39% | 40% | 30% | 35% |
| water level | 171.9 | 169.4 | 165.3 | 164.1 | 156.1 | 145.7 | 150.9 | 148.1 | 156.1 | 172.4 | 174.2 | 173.7 |

**12) Section 3.5 Water eutrophication: In your conclusion, the risk of eutrophication in the tributary bay was highest in the section within 0.5 km of the confluence from May to June. Any facts or references in tributary bays of the TGR that can support your conclusion?**

**Authors' response:** Thank you for your suggestion. We have added the study of Wu et al (2010) to support our conclusion. The following sentences were added in the revised manuscript (Page 34, Line 523 to Line 526).

Wu et al (2010) constantly monitored the eutrophication of the Daning River, a tributary bay of the TGR, and found that algal blooms frequently occurred in the area close to the confluence from March to June, which was similar to the results of the present study.

**13) Line 502- Line 508: You calculated the backwater intrusion time in Section 3.1 and it is a meaningful result. I think you should add this result in the first conclusion.**

**Authors' response:** We have added the result of the backwater intrusion time in the first conclusion in the revised manuscript (Page 36, Line 552).

**14) Line 552 - Line 555: What is the interaction between the main reservoir and the tributary bay? As the tributary is a much smaller water body compared with the main stream, so it's easy to understand the influence of main reservoir on**

**tributary. But can the tributary bay affect the main reservoir conversely? I think there needs more details.**

**Authors' response:** The interaction between the main reservoir and the tributary bay means their hydrodynamics and water environmental characteristics can influence each other. One tributary bay can affect a small section of main reservoir near its confluence, maybe many tributary bays can influence the main reservoir together. A main reservoir's operation may have common influences on its tributary bays. We have added the following sentences in the revised manuscript (Page 37, Line 572 to Line 574).

The operations of the main reservoir may have common influences on the tributary bays, and tributary bays may also influence the main reservoir.

**15) The conclusion part is better to be condensed and proposed some specific conclusion, or some quantify result.**

**Authors' response:** Thank you for your suggestion. We have condensed the conclusion part and put the most quantify results in the conclusion part in the revised manuscript.

**16) Future work: You mentioned some existing measures to improve the environment of tributary bays, can you propose some possible new methods in your future work section?**

**Authors' response:** Thank you for your suggestion. We introduced some possible methods in the future work section. At present, the method of "double nutrient reduction", ecological methods, and manually controlled operation method have been proposed by some scholars. We added this information in the future work section in our manuscript (Page 37 to Page 38, Line 580 to Line 590).

**This paper reports on an investigation of the effects of water level fluctuations in the Three Gorges Reservoir on a tributary bay on the Tangxi River, the focus being on a number of water quality parameters. The study is based on a numerical simulation using the width-averaged vertically two-dimensional model CE-QUAL-W2. It was conducted for the year 2017 and water quality data collected at the Tangxi River Bridge located 18 km upstream from the confluence was used for validation.**

**Authors' response:** Thank you for your commentary. Below we present our responses to each comment.

**Major comments**

**1. While the results address an important problem they are rather limited in scope. The paper could be enhanced, for example, with a discussion of how sensitive the results are to the model forcing, e.g. winds and air temperature. Are the distributions/variations in the water quality parameters driven solely by the water level fluctuations in the reservoir or do the forcings make a contribution?**

**Authors' response:** Thank you for your suggestion. The aim of our paper is to study how a tributary bay was influenced by backwater jacking and intrusions from the main reservoir. The link between the main reservoir and its tributary bay is the hydrodynamic condition, which is mostly affected by the water level fluctuations (Sha et al., 2015). So, we focused on the water level fluctuations in the main reservoir and its influence on the tributary bay in this manuscript.

We used the daily average data of multi-year on winds and air temperature as the boundary in our simulation. We have discussed the sensitivity of our results to the winds and air temperature at a new section in the revised manuscript (Page 35 to Page 36, Line 531 to Line 544).

We also have enhanced our paper in other aspects. For instance, we have added discussions with other tributaries in the results and discussion section. We also have added some references to support our study and added some details to improve the quality of this paper. We hope our efforts to enhance the paper can meet with your approval.

**2. The model validation is limited to comparisons of water quality parameters at a single point: the Tangxi River Bridge. These measurements do not include measurements of currents so there is no validation of the circulation patterns shown in figure 5 or of the two-dimensional distribution of the water quality patterns. This should be commented on and ideally addressed somehow.**

**Authors' response:** Thank you for your comment. We have tried our best to find the fundamental data, but only the data of water quality parameters in Tangxi River Bridge can be got at present. So, we used the data of Tangxi River Bridge to valid the model CE-QUAL-W2. Though the model validation was limited, many scholars have obtained good results by using it. Moreover, this model is mature and has been proved to perform well in simulating the hydrodynamics, water temperature and water quality of reservoirs and lakes. Therefore, we think our results and conclusions are credible. We also have added this information at the end of introduction section (Page 6, Line 106 to Line 111). We hope our explanations for this comment can get your understanding and support.

**3. The title has some grammatical errors: "The hydrodynamic and environmental characteristics of a tributary bay influenced by backwater jacking and intrusions from a main reservoir"**

**Authors' response:** Thank you for your suggestion. We have changed the title to "Hydrodynamic and environmental characteristics of a tributary bay influenced by backwater jacking and intrusions from a main reservoir".

**4. The introduction should include a background discussion on what backwater jacking is and what intrusions from the main reservoir are and the conditions under which they occur. It does not have to be long.**

**Authors' response:** We have added the meanings of backwater jacking and intrusions from the main reservoir and the conditions under which they occur in the revised manuscript (Page 3, Line 52 to Line 55).

**5. The abstract is very long. Seems too long to me.**

**Authors' response:** We have condensed the abstract in the revised manuscript.

**6. Line 14. " ... is the key ...". Is it really true that this is the one an only key to solving eutrophication or is it one more several. I find it hard to believe that it is the only key to solving these problems. Similarly on line 74. Saying "is a key" seems more accurate.**

**Authors' response:** We have changed "is the key" to "is a key" (Page 4, Line 72).

**7. The introduction is very focused on the Three Gorges Reservoir. The paper could be enhanced by adding a discussion of tributary bays in other parts of the world which would help put the work in a wider context.**

**Authors' response:** Thank you for your suggestions. We have tried our best to find the studies of tributary bays in other parts of the world, but the studies were few. However, we have added discussions of other tributary bays of the TGR in the revised manuscript (Page 17, Line 295 to Line 303; Page 22, Line 377 to Line 379; Page 34. Line 523 to Line 526). We hope our efforts to enhance the paper can meet with your approval.

**8. Line 152. Here it is stated that the water density is affected by concentrations of solids (should be 'suspended solids') but equation (6) for the density is a function of temperature only – it does not depend on concentrations of suspended solids. Were these concentrations included in the model somehow? If so this should be explained. If not this should be made clear.**

**Authors' response:** We are sorry that we made a mistake in this statement. The concentrations of suspended solids weren't included in the model. We have revised this sentence in the revised manuscript (Page 9, Line 168 to Line 170) as follows.

Accurate hydrodynamic calculations require accurate water densities. The following equation of state relating the density to the water temperature was used in the model.

**9. What shortwave absorption model was used in this study? A two- or three-band model, or otherwise? With what attenuation coefficients? Fixed or a function of suspended sediments? In parts of the domain (e.g. figure 5) the water is shallow at some times of the year. Does shortwave radiation reach the bottom? If so how is it handled. Does it reflect off the bottom or is that heat absorbed by the bottom potentially creating unstable stratification?**

**Authors' response:** The shortwave absorption we used was according to Bears Law (Thomas and Scott, 2008). The attenuation coefficients in the model include the fraction absorbed at the water surface and the extinction coefficient. The values of them were 0.45 and 0.45 $m^{-1}$ respectively. This information has been added in the revised manuscript (Page 10, Line 191 to Line 194).

As the content of suspended sediments was low in the research area, we didn't consider the suspended sediments in the simulation.

According to our study, the water depth was around 5 m in the upstream from May to September. Most of the shortwave radiation was absorbed by the water, only a small amount of the radiation reached the bottom. Due to the exponential decay of the shortwave radiation, we didn't distinguish the heating after the radiation reached the bottom of the tributary in the simulation. This information has been added in the revised manuscript (Page 10 to Page 11, Line 194 to Line 196).

As for the stratification, the small amount of radiation that reached the bottom of upstream could not cause the vertical convection problem and it had little effect on the stratification. We hope our explanations for this comment can meet with your approval.

**10. I suggest adding a figure showing some of the meteorological forcings: air temperature and wind in particular. The only information on winds and air temperature are the monthly averages in table 1. Why are averages enough? What was the temporal resolution of the forcings used to drive the model: hourly, daily? Were the monthly averaged values used to driving the model? If so why not more frequent values? No diurnal cycle in the forcing? Is the solar radiation in table 1 a combination of long and short wave radiation? These should be reported separately because shortwave radiation penetration penetrates into the water column and longwave radiation does not.**

**Authors' response:** Thank you for your suggestion. Although the meteorological conditions were displayed in the form of monthly average value in the table 1, we used daily average data of multi-year in our simulation. We are sorry that this made you confused.

The diurnal cycle of our simulation last three years. We have added this information in our revised manuscript (Page 13, Line 234).

The solar radiation in table 1 was short wave solar radiation and we have specified this in the revised manuscript. The long wave atmospheric radiation was computed from air temperature and cloudiness.

According to your suggestion, we have replaced the table 1 with a figure of daily average values of meteorological data in the revised manuscript (Page 13, Fig.4) as follows.

[Figure]

**11. Lines 192–193. The percentage error does not seem like a useful metric. A 25% error for a temperature of 4◦ is very different from a 25% error for a temperature of 20◦.**

**Authors' response:** We agree with that the percentage error is not a useful metric, and we have used root mean squared error to reevaluate the model calculation accuracy. We have revised the description of the fitness between simulated values and measured values as follows (Page 11, Line 208 to Line 215).

The difference in T between the simulated value and the measured value was 0.6 - 4.7 ºC, and the root mean squared error was 1.8 ºC. The difference in TP between the simulated value and the measured value was 0.004 - 0.03 mg/L, and the root mean squared error was 0.01 mg/L. The difference in TN between the simulated value and the measured value was 0.02 - 0.26 mg/L, and root mean squared error was 0.16 mg/L. For $NH_3$-N, the difference between the simulated value and the measured value was 0.03 - 0.08 mg/L, the root mean squared error was 0.06 mg/L, and the relative error was greater than 30%.

**12. Figure 5. The left side of the region plotted in each panel varies with month of year. How is this left boundary determined? The ranges of *x* values plotted also**

varies from month to month which makes it a bit difficult to compare results from different months. The panels are also too small. I find them difficult to read. I suggest full page figures with two columns, all using the same range of *x* values. Also, the red curve that is the boundary between Zone 1 and Zone 2 is difficult to see because there is not enough contrast with the colors of the other contour lines. They should be very different. In figures 7 and 9 the curve separating the zones is in black. It would be best to use the same color in all figures. Same comments for other similar figures.

**Authors' response:** Thank you for your suggestions.

The left boundary was determined by the water depth. We set the minimum number of activation layers in the simulation, and the corresponding water depth is 4 m. The simulation stopped when the water depth is less than 4 m, and the left boundary was determined.

If we put the figures into two columns, the figures will become too long and look not good. So, we still arranged the figures into three columns, and we also ensured the accuracy of the figures. We have output clearer figures in the revised manuscript. We have used the same range of x values and uniformed the color of boundary between Zone 1 and Zone 2 in black in the revised manuscript according to your suggestion. We hope our revisions for these figures can meet with your approval (Fig.7, Fig.9, Fig.11 to Fig.14).

**Minor comments**

**1. Line 9. "... by backwater ..." (delete 'the').**

**Authors' response:** We have deleted 'the' in this sentence.

**2. Line 10. "intrusions from the main reservoir". The main reservoir is not intruding into the bay, it is water from the main reservoir which is intruding.**

**Authors' response:** We have corrected this sentence (Page 2, Line 23).

**3. Line 15. "... relevant to the water environment"**

**Authors' response:** We have added deleted this sentence.

**4. Line 17. "... by backwater jacking and intrusions from the ..."**

**Authors' response:** We have deleted this sentence.

**5. Line 19. "... and water quality model ..."**

**Authors' response:** We have added 'water' in front of 'quality model' (Page 5, Line 104).

**6. Line 23. When the water level dropped where? In the main reservoir?**

**Authors' response:** Yes, in the main reservoir. We have revised this sentence in the revised manuscript (Page 1, Line 17 to Line 20) as follows.

The tributary bay was mainly affected by backwater jacking from the main reservoir when the water level of the main reservoir dropped and by intrusions from the main reservoir when the water level of the main reservoir rose.

**7. Line 24. What is a 'quality concentration boundary'?**

**Authors' response:** It is a boundary of the water quality and we have added 'water' in front of 'quality concentration boundary' (Page 1, Line 20).

**8. Line 38. "200 m or even 300 m" is a bit redundant. If dams are 300 m high then it is not necessary to say they are over 200 m high.**

**Authors' response:** We have deleted 'over 200 m' in this sentence.

**9. Line 40. Delete 'However,' and 'the': "These dams block fish .... and change fish communities..."**

**Authors' response:** We have deleted them.

**10. Line 51. "... thus forming water areas ... to lakes known as a tributary bay"**

**Authors' response:** We have revised this sentence in the revised manuscript (Page 3, Line 43 to Line 45) as follows.

Backwater extends to some tributaries after the construction of dammed-river reservoirs, which causes the water depth to increase and the water velocity to slow in these tributaries, thus forming water areas similar to lakes known as a tributary bay.

**11. Line 90. "... to a rise or decline in chlorophyll content depending ...."**

**Authors' response:** We have revised this sentence in the revised manuscript (Page 5,

Line 87 to Line 89) as follows.

A rise in the water level may lead to a rise or decline in the chlorophyll content depending on the water cycle mode in the tributary.

**12. Line 91. Do you mean 'Past studies have paid ...'? If you mean the present study (i.e. this paper) then the grammar is incorrect.**

**Authors' response:** Yes, we mean the past studies. We have changed 'present' to 'previous' (Page 5, Line 89).

**13. Line 96. "by backwater jacking and intrusions from the main ..." This needs fixing in many places.**

**Authors' response:** We have fixed the mistakes in the revised manuscript about this sentence (Page 5, Line 93 to Line 94).

**14. Line 96. The sentence "How the .... tributary bay?" needs to be revised. Perhaps "There are many open questions regarding the functions of these types of systems: How does the operation of the main reservoir affect tributary bays?; How do hydrodynamic forces and the water environment of tributary bays respond to backwater jacking and the intrusion of water from the main reservoir?; What controls the water environment of tributary bays?"**

**Authors' response:** Thank you for your positive and constructive suggestions. We have revised the sentences according to your suggestion (Page 5, Line 95 to Line 98).

**15. Line 103. "... by backwater jacking and intrusions from the TGR ..."**

**Authors' response:** We have revised this sentence (Page 5, Line 100 to Line 101).

**16. Line 106. " and water quality ..."**

**Authors' response:** We have added 'water' in front of 'quality model' (Page 5, Line 104).

**17. Figure 2. The figure caption could be more informative, describing what is shown in each panel.**

**Authors' response:** We have described each panel in the caption of Figure 2. We also have added descriptions in each panel of Figure 1. The new caption of Figure 1 is as follows (Page 7, Line 132 to Line 134).

**Fig. 1.** Research area and hydrologic system of the Tangxi River Basin. (a) Location of the research area relative to China; (b) Location of the research area relative to Chongqing; (c) Hydrologic system of the research area.

**18. Line 131. "The vertical two-dimensional ...W2 solves the width averaged equations and is appropriate from simulating flow in long narrow water bodies. It was adopted for ..."**

**Authors' response:** We have revised this sentence as follows (Page 7 to Page 8, Line 138 to Line 141).

The vertical two-dimensional model CE-QUAL-W2 solves the width averaged equations and is appropriate for simulating flow in long narrow water bodies. This model was adopted for the calculation of the hydrodynamic conditions, water temperature and water quality in the tributary bay.

**19. Line 135. What density current? This is the first mention of a density current.**

**Authors' response:** It's the density-driven current. We mentioned this to explain the model can perform well in backwater intrusion issue.

**20. Line 136. "... results using this ..."**

**Authors' response:** We have deleted 'by' in this sentence.

**21. Line 140. Delete 'listed'.**

**Authors' response:** We have deleted 'listed'.

**22. Lines 156–158. This information should appear directly below equations (1) -(5).**

**Authors' response:** We have moved the explanations of each variable below equations (1) - (5).

**23. Line 183. "... was used to ..."**

**Authors' response:** We have corrected this sentence (Page 11, Line 198).

**24. Line 200. What does "usually exhibits characteristics" mean? I do not**

**understand this.**

**Authors' response:** We are sorry that we missed a word 'complex'. The correct sentence is "…usually exhibits complex characteristics" and we have corrected this in the revised manuscript (Page 11, Line 216).

**25. Line 215. How far away from the tributary bay was the meteorological data collected?**

**Authors' response:** The weather station is about 19.7 km away from the tributary bay. We have added this information in the revised manuscript (Page 12, Line 230).

**26. Line 216. " sources were calculated and included as inputs to the numerical simulations"**

**Authors' response:** We have revised this sentence as follows (Page 12, Line 231 to Line 232).

The pollution loads of point and non-point sources were calculated and included as inputs to the numerical simulations (Table 1).

**27. Line 265. "... nutrient status of ..."**

**Authors' response:** We have corrected this sentence (Page 16, Line 281).

**28. Line 277. Correct grammar.**

**Authors' response:** We have changed 'of' to 'from' in this sentence (Page 17, Line 290).

**29. Line 278. Delete "With the water level fluctuation through the whole year"**

**Authors' response:** We have deleted this.

**30. Line 283. "... length of the backwater ..."**

**Authors' response:** We have corrected this sentence (Page 17, Line 305).

**31. Line 285. "... main reservoir was between 160 and 175 m and the ..."**

**Authors' response:** We have revised this sentence as follows (Page 17, Line 305 to Line 307).

During January to April and October to December, the water level of the main reservoir was between 160 and 175 m and the backwater reached distances of 39.8 - 42.6 km from the confluence simultaneously.

**32. Figure 4 caption. "The relationships among reservoir water level, length ....". The caption should say what the curves are and what the filled in regions are.**

**Authors' response:** Thank you for your suggestion. We have added the legend of this figure in the revised manuscript (Page 18, Fig.6).

**33. Line 302. What is 'water from the tail'?**

**Authors' response:** We have revised this sentence as follows (Page 18, Line 323 to Line 324).

In each month, the upstream water flowed along the surface of the tributary bay or sank to the bottom.

**34. Line 316. What does 'directly flowed to the confluence' mean? Flowed along the surface? This should be clarified. Where is the confluence in the figure?**

**Authors' response:** Yes, we meant the upstream water flowed along the surface. The confluence is the right end of the tributary bay in the figure. We have revised this sentence as follows (Page 19, Line 339 to Line 341).

From July to August, the upstream water of the tributary bay directly flowed to the confluence along the surface layer.

**35. Figure 7. The red contours in the figure should be explained in the caption.**

**Authors' response:** We have added the explanation of the brown contours in the caption. The revised caption of this figure is shown as follows (Page 23, Line 389 to Line 391).

**Fig. 9.** Distribution of water temperature in different months. The black curve in the figure is the boundary between Zone 1 and Zone 2. The brown curves with arrows are streamlines.

**36. Figure 9. Revise caption: "Distribution of COD ...".**

**Authors' response:** We have revised the caption of previous Fig.9 and we also have revised the same errors in the previous Fig.10 - Fig.12.

**37. Line 462. "... was generally higher ..." (it was not higher in every month).**

**Authors' response:** We have added 'generally' in front of 'higher' in this sentence (Page 32, Line 492).

**38. Lines 506. I don't understand what the authors are trying to say here: "brought serve vertical"**

**Authors' response:** We are sorry that this sentence made you confused, and we have deleted this sentence in the revised manuscript.

**39. Line 507. What is meant by "could contrapuntally be proposed"?**

**Authors' response:** We are sorry we used an inappropriate word 'brought' and we have deleted it in the revised manuscript.

[revised manuscript text omitted]

$$\le TLI(\sum) \le 50, \text{mesotrophic}$$

$\quad$ $TLI\,(\textstyle\sum)\,>\,50$, eutrophic

$\quad$ $50\,<\,TLI\,(\textstyle\sum)\,\le\,60$, slightly eutrophic

$\quad$ $60\,<\,TLI\,(\textstyle\sum)\,\le\,70$, moderately eutrophic

$\quad$ $TLI\,(\textstyle\sum)\,>\,70$, severely eutrophic

$\quad$ The formula for calculating the $TLI\,(\textstyle\sum)$ is as follows:

$\quad$ $TLI\,(\textstyle\sum)= \sum_{j=1}^{m} W_j \cdot TLI(j)$ $\hfill$ (9)

where $TLI\,(\textstyle\sum)$ is the comprehensive nutrition index; $W_j$ represents the correlation weight of the nutrition state index of the $j$-th parameter; and $TLI\,(j)$ denotes the nutritional status index of the $j$-th parameter.

$\quad$ Considering chlorophyll-a (*chla*) as the reference parameter, the normalized correlation weight formula of the $j$-th parameter is as follows:

$\quad$ $W_j = \dfrac{r_{ij}^2}{\sum_{j=1}^{m} r_{ij}^2}$ $\hfill$ (10)

where $r_{ij}$ is the correlation coefficient between the $j$-th parameter and the reference parameter *chla* and $m$ represents the number of evaluation parameters.

$\quad$ The correlation coefficients $r_{ij}$ and $r_{ij}^2$ between *chla* and other parameters are shown in Table  2 (Li and Zhang, 1993).

**Table 2**

The correlation coefficients $r_{ij}$ and $r_{ij}^2$ between *chla* and other parameters.

| Parameter | TP | TN | SD | COD$_{Mn}$ |
|---|---|---|---|---|
| $r_{ij}$ | 0.84 | 0.82 | -0.83 | 0.83 |
| $r_{ij}^2$ | 0.7056 | 0.6724 | 0.6889 | 0.6889 |

$\quad$ The calculation formula of the nutritional status index of each parameter is shown as follows:

$\quad$ $TLI(TP) = 10(9.436 + 1.624\ln TP)$ $\hfill$ (11)

$\quad$ $TLI(TN) = 10(5.453 + 1.694\ln TN)$ $\hfill$ (12)

$\quad$ $TLI(SD) = 10(5.118 + 1.94\ln SD)$ $\hfill$ (13)

$\quad$ $TLI(COD_{Mn}) = 10(0.109 + 2.661\ln COD_{Mn})$ $\hfill$ (14)

[revised manuscript text omitted]

---

## Referee Report (RR1)

*Review of*

**"Hydrodynamic and environmental characteristics of a tributary bay influenced by backwater jacking and intrusions from a main reservoir'**

*by X. Li et al.*

This paper has been significantly improved. A few things need to be clarified and corrected so I recommend minor revision.

**Major comments**

1. Lines 59–60. I don't understand what 'the other side is? Do you mean the velocity is higher on the upstream side of the confluence than it is on the downstream side?

2. Line 124. Is the annual flow 14300 $m^3$/s at the mouth of the Yangtze of at the location of the Three Gorges Dam?

3. Line 142. "... by coupling the governing equations ...".

4. Line 143. "The computational domain was divided ..."

5. Lines 152–167. According to figure 2 is appears $\alpha = 0$ so the $gB\sin\alpha$ term in equation (2) should be removed and $\cos\alpha$ should be replaced by 1 in equation (3). On lines 159 it is stated that $z$ is the vertical elevation which, according to Figure 7 increases upward, so the right hand side of (3) should be $-g$. Shouldn't the integrals in (4) be from $h$ to $\eta$ rather than from $\eta$ to $h$?

6. Equation (6). The units for $T_w$ should be given.

7. Line 195. '... we did not distinguish the heating" does not make sense. Some shortwave reaches the bottom. The model must do something with it. From what has been said it sounds like that energy is simply removed. It is not absorbed by the bottom nor is it reflected back into the water column. This should be made clear.

8. Page 11. Is the temperature RMSE for surface water temperature or does it include temperatures at several depths. Similarly for the other quantities.

9. Figure 3. What is plotted here. Surface values? Depth averaged values?

10. Figure 3(b). The meaning of 'Wind direction (E)' is not clear. The direction of the arrows needs to be better explained.

11. Line 295. "Periods of intrusions occurring in other ...."

12. Line 296. "... concentrated during low ... operation and impoundment periods"

13. Lines 301–303. The meaning of this sentence is unclear.

14. Line 326 and below. What is meant by 'one or two flow circulation patterns'. There is one flow circulation pattern in the tributary bay. There can't be two flow circulation patterns at the same time. Do you mean something like 'forming one of two dominant flow circulation patterns'? What are the 'two large counterclockwise circulations' mentions on lines 331–332?

15. Line 349. What is meant by 'flowed through the surface layer'. Flowed along the surface layer?

16. Line 448. Delete 'basically'. What does that word add?

17. Lines 572–574. The meaning of this sentence is not clear.

18. Lines 590–593. The meaning of this sentence is not clear.

---

## Author Response (AR2)

**Dear Editor and Referees,**

On behalf of my co-authors, I would like to thank you for the positive comments and for the opportunity to revise our manuscript entitled "Hydrodynamic and environmental characteristics of a tributary bay influenced by backwater jacking and intrusions from a main reservoir" (ID: hess-2020-63).

We have carefully addressed the comments with point-by-point replies to referee #3 and revised the manuscript accordingly, which we hope will meet with your approval.

In the section below, we have provided detailed responses to the referee's comments and illustrated the corrections made in the paper. The responses are followed by a marked-up manuscript version showing the changes made.

**Responses to the referee #3:**

**This paper has been significantly improved. A few things need to be clarified and corrected so I recommend minor revision.**

**Authors' response:** Thank you for your commentary. We have carefully revised the manuscript according to your suggestions. Below we present our responses to each comment.

**Major comments**

**1. Lines 59–60. I don't understand what 'the other side is? Do you mean the velocity is higher on the upstream side of the confluence than it is on the downstream side?**

**Authors' response:** We are sorry that we made you confused. We have corrected this sentence in the revised manuscript (Page 3, Lines 57-59) as follows.

The horizontal flow velocity near the confluence becomes uneven in the tributary bay, and the flow field distribution tends to gradually change with increasing distance from the confluence (Hu et al., 2013; Yin et al., 2013).

**2. Line 124. Is the annual flow 14300 m³/s at the mouth of the Yangtze of at the location of the Three Gorges Dam?**

**Authors' response:** Yes, the average annual flow of the Yangtze where the Three

Gorges Dam is located is 14300 m$^3$/s. We have made this clear in the revised manuscript (Page 6, Lines 121-123) as follows.

The drainage area of the upper Yangtze River is 527000 km2, and the average annual flow at the location of Three Gorges Dam is 14300 m$^3$/s (Fan, 2007).

**3. Line 142. "... by coupling the governing equations ...".**

**Authors' response:** We have corrected this sentence (Page 8, Line 140).

**4. Line 143. "The computational domain was divided ..."**

**Authors' response:** We have corrected this sentence (Page 8, Line 141).

**5. Lines 152–167. According to figure 2 is appears $\alpha = 0$ so the $gB\sin\alpha$ term in equation (2) should be removed and $cos\ \alpha$ should be replaced by 1 in equation (3). On lines 159 it is stated that $z$ is the vertical elevation which, according to Figure 7 increases upward, so the right hand side of (3) should be $-g$. Shouldn't the integrals in (4) be from $h$ to $\eta$ rather than from $\eta$ to $h$?**

**Authors' response:** We are sorry we made this mistake. We have corrected it in the revised manuscript according to your comment (Page 8 and Page 9, Line 152 and Line 154).

**6. Equation (6). The unit for $T_w$ should be given.**

**Authors' response:** The unit for $T_w$ is °C, and we have given it in the revised manuscript (Page 9, Line 171).

**7. Line 195. '... we did not distinguish the heating" does not make sense. Some shortwave reaches the bottom. The model must do something with it. From what has been said it sounds like that energy is simply removed. It is not absorbed by the bottom nor is it reflected back into the water column. This should be made clear.**

**Authors' response:** We are sorry that this sentence was not expressed clearly. We have replaced the original sentence with the following sentence in the revised manuscript (Pages 10, Lines 192-194).

Most of the shortwave radiation was absorbed by the water, and the other small part of shortwave radiation reaching the bottom was considered as being reflected back into the water column.

**8. Page 11. Is the temperature RMSE for surface water temperature or does it include temperatures at several depths. Similarly for the other quantities.**

**Authors' response:** All the quantities used to calculate RMSE were average values at 0-5 m depth and we have made it clear in the revised manuscript (Page 11, Lines 198-199).

**9. Figure 3. What is plotted here. Surface values? Depth averaged values?**

**Authors' response:** We are sorry that we did not make it clear. Average values of surface 5 m are plotted in Figure 3 and we added the following sentence in the revised manuscript (Page 11, Lines 198-199). We also added this information in the caption of Figure 3.

Average simulated values at 0-5 m depth were used to compare with the measured values.

**Fig. 3.** Comparison between the average simulated values at 0-5 m depth and measured values at the Tangxi River Bridge in each month. (a) Comparison of water temperature; (b) Comparison of ammonia nitrogen; (c) Comparison of total phosphorus; (d) Comparison of total nitrogen.

**10. Figure 3(b). The meaning of 'Wind direction (E)' is not clear. The direction of the arrows needs to be better explained.**

**Authors' response:** Thank you for your suggestion. We have added the explanation at the caption of Figure 3. The new figure 3 in the revised manuscript is shown as follows.

[Figure]

**Fig. 4.** Meteorological conditions. (a) Daily average multi-year values of air temperature, humidity and cloudiness and (b) daily average multi-year values of wind conditions and shortwave radiation. Arrows in (b) indicate the wind direction, and the arrow upward is defined as the direction of due north.

**11. Line 295. "Periods of intrusions occurring in other ...."**

**Authors' response:** We have corrected this sentence according to your suggestion (Page 17, Line 295).

**12. Line 296. "... concentrated during low ... operation and impoundment periods"**

**Authors' response:** We have corrected this sentence (Page 17, Lines 296-297).

**13. Lines 301–303. The meaning of this sentence is unclear.**

**Authors' response:** We have corrected this sentence in the revised manuscript (Page 17, Lines 301-303) as follows.

The results of this study and previous studies indicated that the backwater intrusions showed obvious seasonal changes and the intrusion time was almost the same.

**14. Line 326 and below. What is meant by 'one or two flow circulation patterns'. There is one flow circulation pattern in the tributary bay. There can't be two flow circulation patterns at the same time. Do you mean something like 'forming one of two dominant flow circulation patterns'? What are the 'two large counterclockwise circulations' mentions on lines 331–332?**

**Authors' response:** What we wanted to express was 'one or two flow circulations'. We used an inappropriate word 'patterns' and it made you confused. We have rewritten this sentence as follows (Page 18, Lines 325-326).

The backwater from the main reservoir entered the confluence at different depths simultaneously, forming one or two flow circulations in the tributary bay.

We are sorry that we made a mistake on lines 331-332. It should be 'a' but not 'two' and we have corrected it in the revised manuscript (Page 19, Line 330).

**15. Line 349. What is meant by 'flowed through the surface layer'. Flowed along the surface layer?**

**Authors' response:** We have corrected this sentence according to your suggestion (Page 19, Line 348).

**16. Line 448. Delete 'basically'. What does that word add?**

**Authors' response:** We have deleted it at line 446 in our revised manuscript. We have also corrected other two same errors at line 21 and line 563.

**17. Lines 572–574. The meaning of this sentence is not clear.**

**Authors' response:** We have rewritten this sentence in the revised manuscript (Page 36, Lines 570-571) as follows.

The tributary bays may also influence the main reservoir.

**18. Lines 590–593. The meaning of this sentence is not clear.**

**Authors' response:** We have rewritten this sentence in the revised manuscript (Page 37, Lines 587-589) as follows.

[revised manuscript text omitted]